


# 1 Atmospheric stability from microwave radiometer

# 2 observations for on/offshore wind energy applications

Domenico Cimini[1,2], Rémi Gandoin[3], Stephanie Fiedler[4,a], Claudia Acquistapace[5], Andrea
Balotti[2], Sabrina Gentile[1,2], Edoardo Geraldi[1], Christine Knist[6], Pauline Martinet[7], Saverio T.
Nilo[1], Giandomenico Pace[8], Bernhard Pospichal[5], Filomena Romano[1]
[1] CNR-IMAA, C.da S.Loja, Potenza, 85100, Italy
[2] CETEMPS, University of L'Aquila, L'Aquila, 67100, Italy
[3] C2Wind, Fredericia, 7000, Danmark
[4] University of Cologne, Cologne, Germany
[5] Institute for Geophysics and Meteorology, University of Cologne, 50969 Cologne, Germany
[6] DWD, Meteorological Observatory Lindenberg – Richard Aßmann Observatory, Tauche OT Lindenberg, 15848,
Germany
[7] CNRM, Université de Toulouse, Météo-France, CNRS, Toulouse, France.
[8] ENEA, Observations and Measurements for Environment and Climate Laboratory, Rome, 00123, Italy
[a] Now at Institute of Environmental Physics, University Heidelberg, D-69120 Heidelberg, Germany
*Correspondence to*: Domenico Cimini (domenico.cimini@imaa.cnr.it)
**Abstract.** Atmospheric stability controls the evolution of wind turbine wakes, and thus the yield and performance
of wind parks. For estimations of wind park power output and for improving analyses of wind park wakes, crucial
parameters were found to be profiles of atmospheric temperature and stability metrics. Atmospheric temperature
profiles are available from numerical weather prediction (NWP) models or are measured in-situ by balloon-borne
sensors, but can also be estimated from the ground using radiometric observations. This paper reviews the stability
metrics useful for monitoring wind park performances and provides a quantitative assessment of the value of NWP
model data for estimating these metrics. This paper also extends previous work, quantifying the performances of
microwave radiometer (MWR) observations to estimate stability metrics from surface-based observations in three
climatological conditions (marine, continental, and polar) and with different instrument types, either situated on
land or ocean. Two NWP systems (DOWA and NEWA) have been evaluated against temperature profiles
measured by offshore met masts in the 30-100 m layer from the surface. Systematic differences are ~0.3-0.5 K,
with no clear dependence on the stability class. Conversely, both models show larger random differences in stable
than in unstable conditions. Root-mean-square (RMS) differences were within 1 K for DOWA, while it exceeded
2 K for NEWA in very stable conditions. For temperature gradients in the 50-100 m vertical layer, the mean
absolute error (MAE) was ~3.4-4.2 K/km, with 5.8-8.4 RMS, and 0.7-0.8 correlation. From the six datasets of
MWR and radiosonde observations considered here, temperature profiles mostly agree within ~0.5 K near the
surface increasing to ~1.5 K at 2 km. Substantial differences are found between MWR performances in retrieving
temperature and potential temperature gradients (50-300 m) onshore and offshore. Onshore, potential temperature
gradients agree with 2.1-3.4 K/km MAE and 0.7-0.9 correlation. Offshore, both MAE (0.9-1.9 K/km) and
correlation (0.3-0.4) are relatively lower, although performances tend to improve using elevation scanning
retrievals. Considering all the datasets, reported MAE are 0.9-3.4 K/km, while RMS are 1.2-5.1 K/km. Thus, the
uncertainty of MWR for temperature and potential temperature gradients in the 50-300 m vertical layer is ~0.5-
4.3 K/km. The relatively lower performances off-shore may be attributed to the training of the inversion method,
which may under-represent the peculiar off-shore conditions, and the ship movements, which can impact low-
elevation observations. These considerations suggest that appropriate dedicated training and elevation scanning
with ship movement compensation may be required for MWR to better catch potential temperature gradients
typical of offshore conditions.

## 1 Introduction

Stability is a characteristic of how a system reacts to small disturbances. If the disturbance is damped, the system
is considered to be stable. If the disturbance causes an amplifying response, the system is unstable (Stull, 2017).
Atmospheric stability is a measure of the atmospheric state which determines whether air will tend to rise or sink
(Spiridonov & Ćurić, 2021). In simple words, a layer is considered as stable when vertical motion is suppressed,
and as unstable (or convective), when vertical motion is enhanced (Stull, 2012). Stability conditions are often
mainly driven by the balance between momentum and heat fluxes close to the surface and can be described by
similarity laws (Gryning er al, 2007). However, there are conditions under which the characterisation of stability
requires detailed information of the atmospheric boundary layer (ABL) across height, for instance when warm air
is advected aloft over a colder surface. In fact, the buoyancy ($B$), that is the acceleration of an air parcel after a
certain displacement over height ($\Delta z$) is proportional to the atmospheric potential temperature ($\theta$) and its vertical
gradient ($\frac{d\theta}{dz}$), as:
$$B = -\frac{g\Delta z}{\theta}\frac{d\theta}{dz}$$
$$(1)$$
where $g$ is the gravitational acceleration and $\theta$ is defined through air temperature ($T$) and pressure ($P$) as (Stull,

60  2012):

$$\theta = T\left(\frac{P_0}{P}\right)^{R/c_p}$$
$$(2)$$
with $P_0$ as reference pressure (e.g., 1000 hPa), and $R/c_p$ the ratio between the gas constant and the specific heat
capacity at a constant pressure for air. If the parcel is moved up ($\Delta z > 0$) and $\frac{d\theta}{dz} < 0$, the buoyancy tends to lift the
parcel further ($B > 0$, instability); conversely, if $\frac{d\theta}{dz} > 0$ the buoyancy moves the parcel back towards its original





location (*B<0*, stability). Atmospheric stability is relevant for meteorological processes and applications,
including air quality, and renewable energy yield. In particular, atmospheric stability is relevant for the prediction
of vertical wind shear (larger during stable conditions) and turbulence (larger during unstable conditions). Wind
turbine rotors span a relatively large range of elevations (between approx 23 to 250 m ASL for a modern turbine),
so the thermodynamic conditions in the lowest 300 m are the most relevant for this application. In particular,
atmospheric stability has a major impact on the characteristics of wind turbine wakes and thus on the yield and
performance of offshore wind parks (Hansen et al, 2012). However, simple approaches for defining stability, e.g.,
using surface layer stability metrics such as the Obukhov length (Obukhov, 1971; Foken, 2006) or the temperature
difference between the sea surface and the atmosphere at one particular altitude, are not always suitable for
describing stability conditions and wake development. For the estimation of wind park power output and for
improving analyses of offshore wind park wakes, atmospheric temperature profiles and stability metrics were
found to be crucial parameters. In fact, improved characterisation of wind farm output can be produced if the
boundary layer stability is considered, indicating the need for temperature measurements at separate heights
(Vanderwende and Lundquist, 2012). Different power curves shall be calculated for different stability conditions,
leading to more accurate and reliable performances of energy production calculations (St. Martin et al., 2016).
For example, for a wind energy farm in a coastal region, Perez et al. (2023) reported that unstable atmospheric
conditions deliver up to 8% more power than stable conditions, while neutral conditions deliver up to 9% more
energy than stable conditions. As a small percent difference leads to a large deviation in cost for both operators
and manufacturers, calculating different power curves for different atmospheric conditions lowers the financial
risks for both operators and manufacturers (St. Martin et al., 2016). In particular, temperature inversions are
important, which may occur above, below, and within the wind turbine rotor area. These conditions would affect
the wake development in different ways, e.g., (*i*) decoupling the wake from the surface or (*ii*) preventing the wake
vertical spreading for inversions below/above the rotor area, respectively (Platis et al., 2020).
Atmospheric temperature profiles can be measured in situ by sensors located on instrumented towers, drones, and
balloons. Instrumented towers have the advantage of providing temperature profiles nearly continuously in time.
However, the costs for their installation and maintenance are quite demanding, and particularly impractical on
offshore platforms, resulting in limited deployment (up to ~100 m height, to our knowledge). Also drones have
limited range in altitude with about 120 m in US and Europe, unless special waiver by corresponding aviation
safety agencies (Pinto et al., 2021; Hervo et al., 2023), and in addition their use requires attended service.
Conversely, sondes attached to balloons, referred to as radiosondes, can nowadays be launched by automatic
stations (Madonna et al., 2021) and usually reach elevations well above the ABL (25 km altitude or more). Each
radiosonde measures one instantaneous and vertically high-resolution profile of atmospheric temperature,
humidity, wind speed and direction. However, the cost of a radiosonde launch is such that they are typically





launched once or twice a day, except at major atmospheric observatories run by meteorological services that have
up to four radiosondes per day or during field campaigns with a radiosondes program to meet research needs.
Remote sensing technology has the potential to overcome some of the limits of in-situ measurements. Ground-
based measurements of atmospheric temperature and humidity profiles are possible using microwave radiometers
(MWR, Cimini et al., 2006), infrared spectrometers (IRS, Feltz et al., 2003), and radio-acoustic sounding systems
(RASS, Bianco et al., 2017). These remote sensing systems provide unattended operations and high temporal
resolution (order of minutes) measurements that are used for a range of applications, including operational
meteorology (Cimini et al., 2015; Shrestha et al., 2021), atmospheric study processes (Martinet et al., 2017;
Martinet et al., 2020; Wagner et al., 2022), and weather forecast (Caumont et al., 2016; Lin et al., 2023; Cao et
al., 2023). Conversely, atmospheric thermodynamic profilers have not been exploited extensively for wind energy
applications, despite the general recognition of the importance of temperature profiles and atmospheric stability
regimes for the characterization of wind energy production (Vanderwende and Lundquist, 2012; St. Martin et al.,
2016; Perez et al., 2023). Ongoing efforts in this direction include the series of on/offshore field campaigns
performed within the Wind Forecast Improvement Projects (WFIP, Wilczak et al., 2015; Shaw et al., 2019;
Wilczak et al., 2019). Although the uncertainty requirements for atmospheric stability measurements to serve
wind energy applications have not been assessed yet, it is useful to assess the accuracy currently achievable by
remote sensing thermodynamic profilers. To this end, Bianco et al. (2017) assessed the accuracy of MWR and
RASS in light of onshore wind energy applications. This study proved that these remote-sensing instruments can
provide accurate information on atmospheric stability conditions in the ABL, with 0.87-0.95 correlation between
temperature lapse rate in the 50-300 m range as measured by a MWR and tower sensors (note that here and
throughout this paper correlation is evaluated with the Pearson's linear correlation coefficient, not to be confused
with the determination coefficient $R^2$ used elsewhere, e.g., by Bianco et al., 2017). Combining this with the need
for temperature gradients for onshore and offshore wind energy (e.g., Platis et al. 2020; Perez et al., 2023), it
seems natural to extend the investigation of MWR performances to other environmental conditions.
Building on these premises, the Carbon Trust Offshore Wind Accelerator (OWA) funded the Radiometry and
Atmospheric Profiling (RAP) scoping study. RAP aimed at assessing existing MWR technology and its
performances for atmospheric profiling and stability measurements. This paper presents the main outcomes of the
RAP project. Section 2 presents a review of capabilities from numerical weather prediction (NWP) modelling
systems (hereafter: NWP models), which represent the default option in the absence of measurement data. Section
3 briefly introduces MWR technology currently available and the datasets exploited for this analysis. Section 4
presents the validation of temperature gradients measured by MWR units with respect to reference radiosonde
data. Section 5 presents a summary, conclusions, and plans for dedicated onshore and offshore field campaigns.

## 2 Validity assessment of NWP datasets

As part of the RAP project, the validity of NWP models for assessing atmospheric stability for the purposes of offshore wind engineering was investigated. The following model datasets were used: (*i*) ERA5 from the ECMWF (Hersbach et al., 2020) obtained via the Copernicus Climate Change Service (C3S, 2021), (*ii*) the New European Wind Atlas (Lundtang Petersen, 2014; NEWA, 2021), and (*iii*) the Dutch Offshore Wind Atlas (Wijnant et al., 2019; DOWA, 2021). NEWA and DOWA have been produced using two different mesoscale NWP models, and both use ERA5 as input. Measurement data came from the FINO1, FINO2 and FINO3 met masts, via the German Federal Maritime and Hydrographic Agency (https://www.bsh.de/EN/), and from the IJmuiden met mast as well as floating lidar measurements in the Southern North Sea, via The Netherlands Enterprise Agency (https://english.rvo.nl/).

### 2.1 Surface stability metrics

For characterising atmospheric stability in the surface layer, pre-existing validation studies have been used (i.e., Peña et al., 2008; Peña and Hahmann, 2011; Sathe et al, 2011). In order to validate the wind speed profile analytical models proposed originally by Gryning et al. (2007), the focus was set on the Obukhov length ($L$):

$$L = \frac{-u_{*0}^3}{\kappa(g/T)\overline{w'T'}_0}$$

$$(3)$$

where $u_{*0}$ and $\overline{w'T'}_0$ are respectively the friction velocity and kinematic heat flux at the surface, $\kappa$ is the von Karman constant ($\approx 0.4$), $T$ the temperature, and $g/T$ the buoyancy parameter. The NEWA Obukhov length time series are readily available, while for ERA5 it was derived from the single levels datasets using two methods: firstly using the turbulent fluxes, and secondly computing the bulk Richardson number ($Ri_b$) from sea surface temperature, air temperature and wind speed at 2 and 10 m ASL, respectively, and relating $Ri_b$ to the dimensionless stability parameter $z/L$ (where $z$ is the height above ground level), i.e.:

$$\frac{z}{L} = C_1 Ri_b$$

$$(4)$$

$$\frac{z}{L} = \frac{C_2 Ri_b}{1 - C_3 Ri_b}$$

$$(5)$$

for unstable and stable conditions, respectively (Peña et al., 2008). The values of $C$ constants are adopted from Grachev and Fairall (1997): $C_1 = C_2 \approx 10$ and $C_3 \approx 5$. Similarly, the Obukhov length was derived from measurements, i.e., the HKZA floating lidar dataset (de Montera et al., 2022) using the same method (via the bulk Richardson number) mentioned above for ERA5: the 10 m ASL wind speed was derived from the 4 m ASL sonic anemometer and three smallest lidar elevations at 20, 30 and 50 m ASL. The results from the models and



measurements are compared in Figure 1. Overall, the best match between model data and measurements is
observed for ERA5 datasets computed using the fluxes for unstable conditions (i.e. $10/L < -0.03$). For stable
conditions ($10/L > 0.03$), the best match is observed when using the bulk Richardson number-derived ERA5
time series. These results confirm that when the main drivers of atmospheric stability (i.e., air- and surface
temperature difference, wind speed) are correctly characterised by the bulk formulations used in NWP models,
the modelled Obukhov length time series compare well - in an average sense - to those derived from
measurements. This implies that such results are hardly generalisable, that is: the user of model datasets should
check, across the region of interest, the validity of these key variables. This can for instance be done using buoy
measurements, where available.


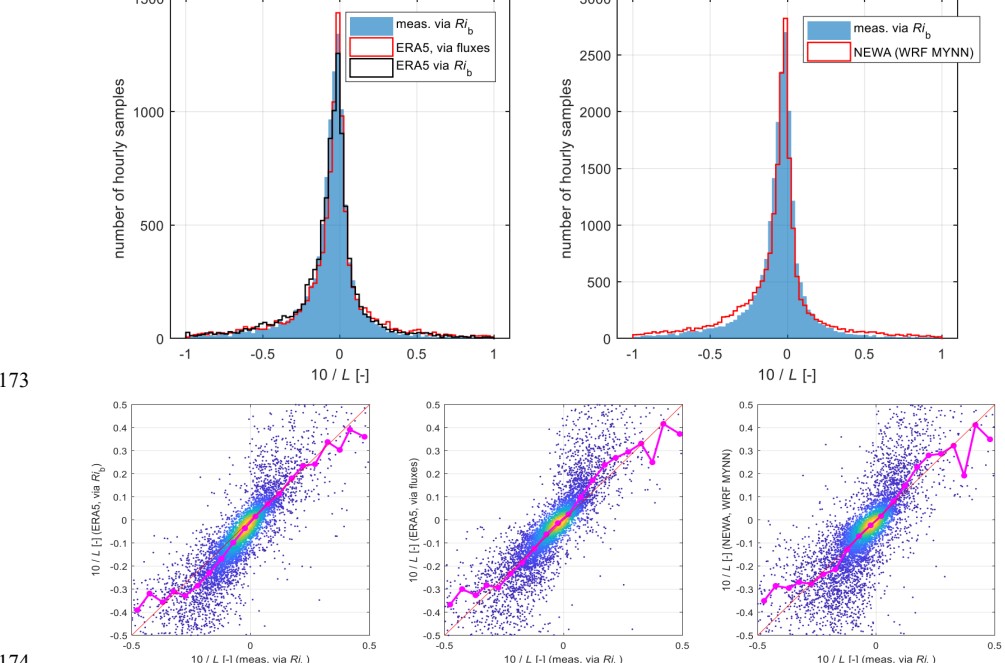


**Figure 1.** Top: Histograms of the dimensionless stability parameter $z/L$ (where $z = 10$ m MSL and $L$ is the Obukhov length).
Bottom: Comparison between measured ($Ri_b$-based) values and modelled values (see y-axes) of $z / L$, at the HKZA FLS
location, and using ERA5 and NEWA model data, for 10 m MSL wind speeds larger than 10 m/s. The measurements have
been averaged to hourly values.





Practitioners are primarily interested in how these modelled Obukhov length time series can improve wind-related
analyses. Two examples are provided in Figure 2 and Figure 3; they both use ERA5-derived Obukhov length time
series derived from fluxes. The first example focuses on turbulence intensities (*TI*), i.e., the ratio of the root-mean-
square of the eddy velocity to the mean wind speed, and mean horizontal wind speed (*WS*). Figure 2 shows how
measured *TI* and *WS* spectra vary with the atmospheric stability class indicated by the modelled Obukhov length:
as classically reported in the literature, turbulence intensities are smaller in stable ($z/L \geq \sim 0.1$) than in unstable
($z/L \leq \sim -0.1$) conditions. In addition, the *WS* spectra progressively increase as conditions shift from stable to
neutral to unstable. The second example shows how the set of analytical expressions proposed by Gryning et al.
(2007) and the method outlined in Peña et al. (2008) compare with simpler, surface-layer expressions, such as the
Monin-Obukhov Similarity Theory (MOST). Note that the expressions from Gryning et al. (2007) basically form
an extension of the MOST above the surface layer (SL). Figure 3 confirms that MOST predicts well the wind
speed profile in neutral and unstable conditions, while it significantly overpredicts the measurements above 30 m
in very stable conditions. This is due to the influence of other scaling parameters such as the boundary-layer
height, which is not accounted for in surface-layer scaling. Figure 3 also corroborates the findings from Peña et
al. (2008), showing that accounting for the effect of the boundary-layer height in stable conditions is essential to
better capture the wind speed above 30 m with respect to MOST, correcting the overprediction up to the boundary-
layer height. This improvement is due to better modelling of characteristic length scales of the turbulent eddies
for the ABL layers located above the surface layer, especially in stable conditions when the surface layer is very
shallow (i.e., less than 100 m in depth).




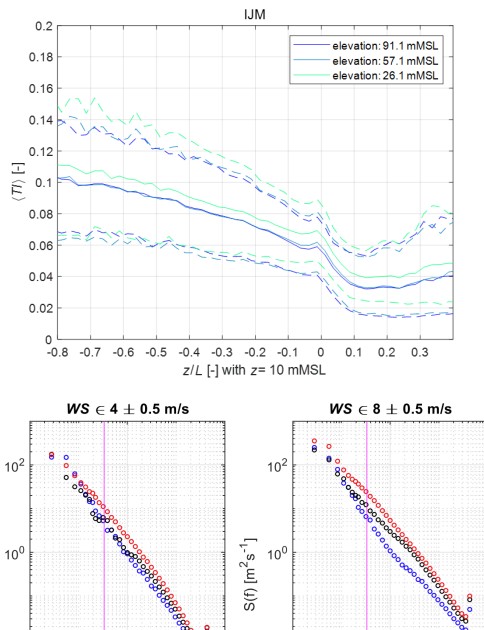


**Figure 2.** Left: Dependence of the turbulence intensity (TI) on the atmospheric stability, for the IJmuiden met mast dataset. Here, the stability is expressed on the x-axis using the Monin-Obukhov length L and the ratio z/L with z = 10 m MSL. Different line colours indicate TI measured at different measurement heights. The full lines are mean values, the dashed lines are 10- and 90-percent quantiles. Two right plots: mean hourly power spectra measured at the top of the IJmuiden met mast (91.1 m MSL), for various stability classes (blue: stable, red: unstable, black: neutral), and two wind speed bins (8 and 12 m/s, respectively). The vertical magenta lines indicate 3.3 mHz frequency, corresponding to 5-minute interval (1/300s).

208



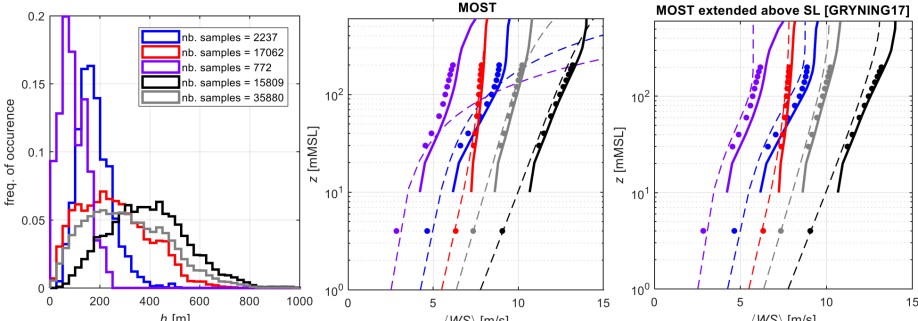

**Figure 3.** Left: histograms of the boundary layer height ($h$) as defined in Gryning et al. (2007). Different colours correspond to different stability classes: purple is very stable, blue is stable, red is unstable, black is near-neutral and neutral, and grey shows all data. Center: profiles of wind speed; dots are measurements from the HKZA floating lidar, full lines are from the DOWA dataset, while the dashed lines are from the MOST. Right: Same as in the centre, but for the MOST SL-extended model from Gryning et al. (2007).

It is concluded that for offshore areas during cases when the main drivers of atmospheric stability are correctly characterised by NWP models, these results can provide wind energy practitioners with valid (in an average sense) Obukhov length time series which can be used for a range of analyses, including estimates of turbulence and wind shear. However, in specific cases, the simulated profiles need to be carefully assessed with observations, since the wind speed profiles and hence the vertical shear and associated turbulence characterization may not be sufficiently accurate. This is a long-standing limitation especially for stably stratified boundary layers (Sandu et al., 2013).

### 2.2 Temperature profiles across the ABL

The validity of NWP model data to characterise the air temperature profile in different stability conditions was assessed using air temperature measurements from tower sensors located approximately from 30 to 100 m ASL. Only DOWA and NEWA data were available at the same elevations as the measurements, while the ERA5 provides only few samples at these elevations. Tower measurements and model data have been divided in five classes of stability conditions: very unstable ($10/L < -0.1$), unstable ($-0.1 \leq 10/L < -0.05$), neutral ($-0.05 \leq 10/L < 0.05$), stable ($0.05 \leq 10/L < 0.1$), and very stable ($10/L \geq 0.2$). Figure 4 shows mean temperature profiles from measurements and model data and their mean and RMS differences in those five classes. DOWA and NEWA models give similar results, providing temperature profiles close to measurements in average for all stability conditions. Mean differences range from ~0.3 to 0.5 K, with no clear pattern with respect to stability class. Conversely, both DOWA and NEWA models show increased RMS in stable conditions with respect to unstable conditions, with minimum RMS in neutral conditions. DOWA seems to perform better (RMS within 1





K throughout the 30-100 m range) than NEWA, especially in very stable conditions (RMS up to 2.2 K). To
measure the NWP overall performances in modelling atmospheric stability, one may look at the performances in
predicting the vertical gradient of temperature ($\frac{dT}{dz}$). In fact, recalling Eq.(1), stability directly depends upon the
vertical gradient of potential temperature ($\frac{d\theta}{dz}$), which is well correlated with $\frac{dT}{dz}$. This is shown in Figure 5,
reporting the scatter of $\frac{dT}{dz}$ between 50 and 100 m ASL as modelled by the DOWA and NEWA datasets and
measured by the tower sensors at the FINO1 and FINO3 platforms. Data points are quite scattered, with model
data covering a range (~100 K/km) lower than measurements (~200 K/km). As for the temperature profiles, the
DOWA dataset performs better than the NEWA, in terms of mean absolute error (MAE, 3.4 to 4.0 K/km for
DOWA, 3.5 to 4.2 K/km for NEWA), RMS (5.8 to 7.3 K/km for DOWA, 6.4 to 8.4 K/km for NEWA), and
correlation (0.77 to 0.80 for DOWA, 0.70 to 0.71 for NEWA).


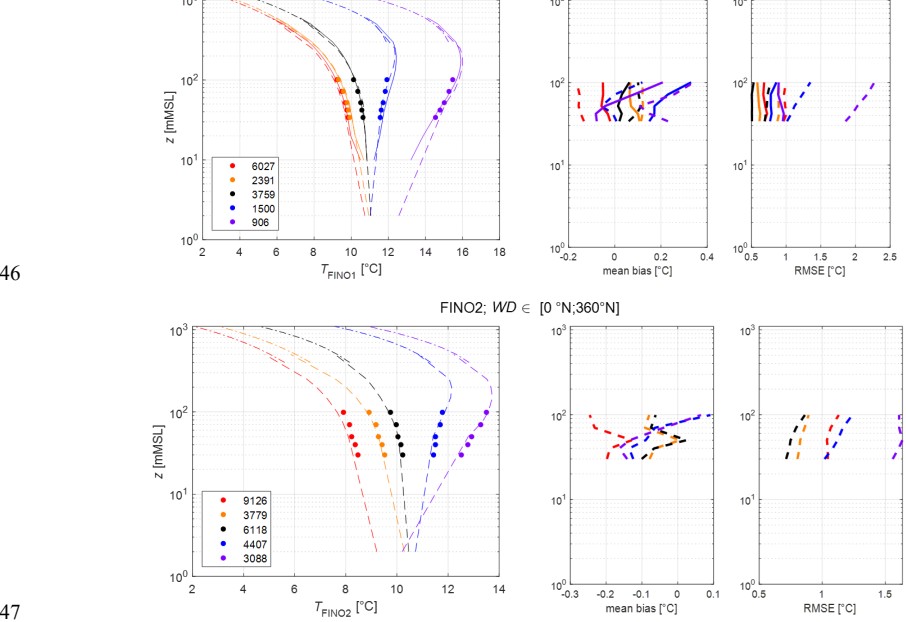




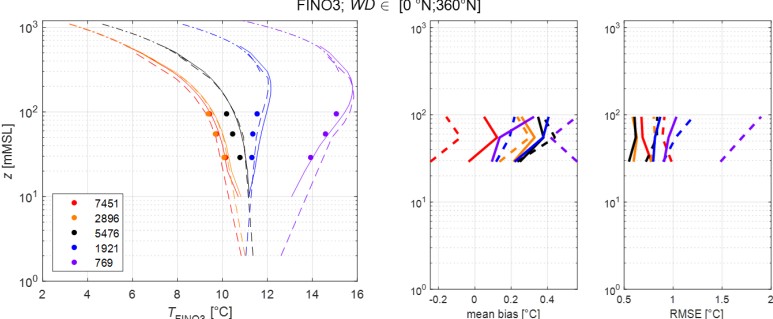

**Figure 4.** Left: Temperature profiles from measurements (dots) and model data (full lines: DOWA, dashed lines: NEWA, dash dotted lines: ERA5) at three measurement locations: FINO1 (top), FINO2 (middle), and FINO3 (bottom). Center: mean model minus measurement temperature differences. Right: temperature RMS differences. Colours indicate stability class: very unstable (red), near-neutral and unstable (orange), neutral (black), near-neutral and stable (blue) and very stable (purple). DOWA data are not shown in the middle panels as DOWA's domain does not cover FINO2 area.

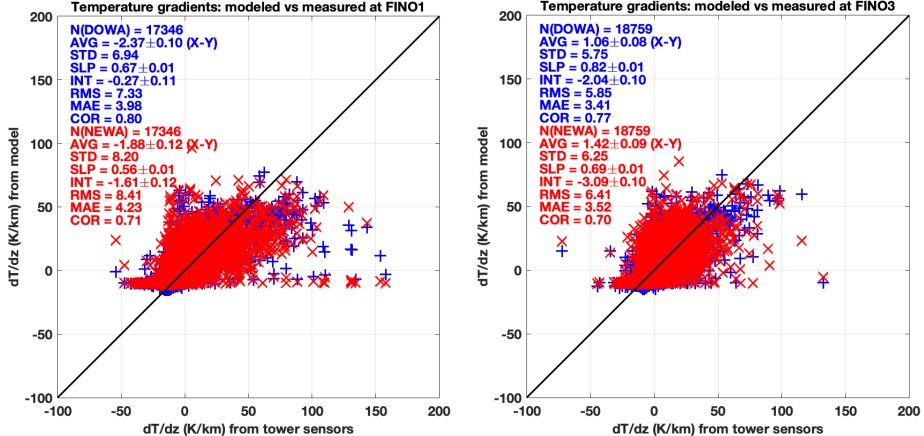

**Figure 5.** Scatter plots of atmospheric temperature lapse rate (~50–100 m) from tower measurements at FINO1 (left) and FINO3 (right) and model data (DOWA: blue crosses; NEWA: red Xs). N indicates the sample size, AVG the average difference (± 95% confidence interval), STD the standard deviation, SLP and INT the slope and intercept of a linear fit (± 95% confidence interval), RMS the root-mean-square, MAE the mean absolute error, and COR the correlation coefficient $R$. Units for AVG, STD, RMS, and MAE are in K/km.



### 2.3 Conditions for difficult stability characterization

The results from Sections 2.1 and 2.2 show that surface stability metric can suffice for a number of analyses, where the model results are validated in an average sense (mean- wind speed or turbulence intensity, for instance). Other purposes require investigating short-lasted events, characterised by different stability conditions at the surface compared with higher elevations. This is for instance the case for the interpretation of wind maps from synthetic aperture radar (SAR) observations or in-situ profile measurements from uncrewed aircraft systems (UAS) as in, e.g., the WInd PArk Far Fields (WIPAFF) project where both of these measurement types were used (Platis et al., 2020). An illustrative example is provided in Figure 6, where SAR-derived 10 mASL wind speeds are plotted over an area covering the Belgian offshore wind farm cluster. Figure 6 also shows the SAR-derived wind speeds across the cluster, as well as mean wind speed profiles measured at the BWFZ01 location together with model data (which do not include the wind farms), and the vertical temperature profiles from ERA5 and DOWA NWP models. The situation seems to correspond, according to the DOWA and ERA5 data, to neutral conditions at the surface, with a stable inversion cap at ~150 mASL. The SAR-derived winds show that the wakes from the Belgian cluster extend over a long distance (tens of kilometres), and the reason is likely the very steep gradient in potential temperature (27 K/km between 120 and 150 mMSL), capping the lowest (neutral in this case) layer of the atmosphere. This at least what the DOWA model indicates, as there are no air temperature measurements which can confirm this. In any case, the observed, and modelled surface stability metrics indicate unstable to neutral conditions at the surface; this would be an incorrect way to characterise the wind flow controlling the wind farm wake, which is very likely located in a stable layer.



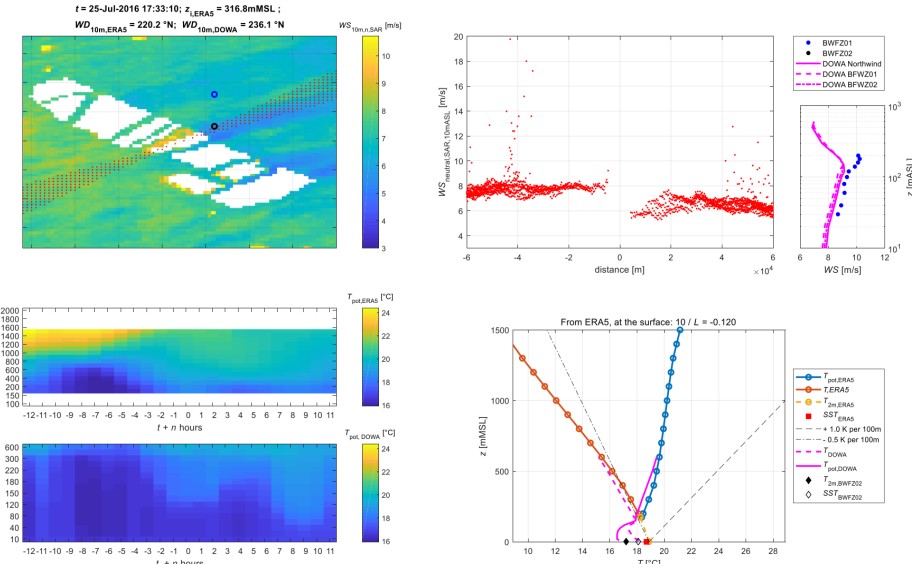


**Figure 6.** Example of a long wake episode across an offshore wind cluster in the Belgian North Sea on July 25th 2016.
Clockwise from top-left: (a) SAR-derived 10 mASL wind speeds mapped over an area covering the Belgian offshore wind
farm cluster (white areas indicate lease areas; red dots indicates the location of wind speeds reported in panel (b); blue circles
indicate the location of two floating lidars, BWF01 and BWF02). (b) SAR-derived wind speeds crossing the offshore cluster
(from -60 to 60 km distance, where 0 indicates the centre of the cluster). (c) Wind speed profiles from floating lidar
measurements and DOWA model at the two downwind sites shown in panel (a). (d) Temperature and potential temperature
profiles from NWP models ERA5 and DOWA at the time of the SAR image. The red square indicates the sea surface
temperature (SST) from ERA5, while the diamonds indicate SST (empty) and 2-m temperature (filled) from measurements at
BWFZ02. Dashed and dot-dashed grey lines indicate +1K and -0.5K per 100 m gradients. The estimated Obukhov length at
the surface is reported, indicating unstable to neutral conditions (10/L=-0.120). (e) Vertical temperature profiles from NWP
models from 10 to 600 m (DOWA) and 160 to 1600 m (ERA5) in the 12 hours before and after the SAR image.

To further investigate the uncertainty associated with the NWP models for such transient flow events, air
temperature data from the NEWA dataset have been compared with measurements from the WIPAFF project
(Bärfuss et al, 2019). For each of the WIPAFF flights, the NEWA air temperature data have been spatially and
temporally interpolated at the UAS locations (down-sampled, from the original dataset). Figures such as Figure 7
have been produced for each WIPAFF flight and are provided as supplement material. The plots indicate the need
for temperature measurements above 100 mMSL, as they suggest that such measurements could help understand
whether such important phenomena for wind farm wake modelling as temperature inversions are well captured



by mesoscale models when they occur above 100 mMSL, where measurements are often not available. Such a
need may be satisfied by nearly continuous observations from a microwave radiometer profiler. The ability to
profile atmospheric temperature continuously within the first 2 km and to provide potential temperature gradients
in the vertical range of wind turbine rotors is assessed in the next Section.

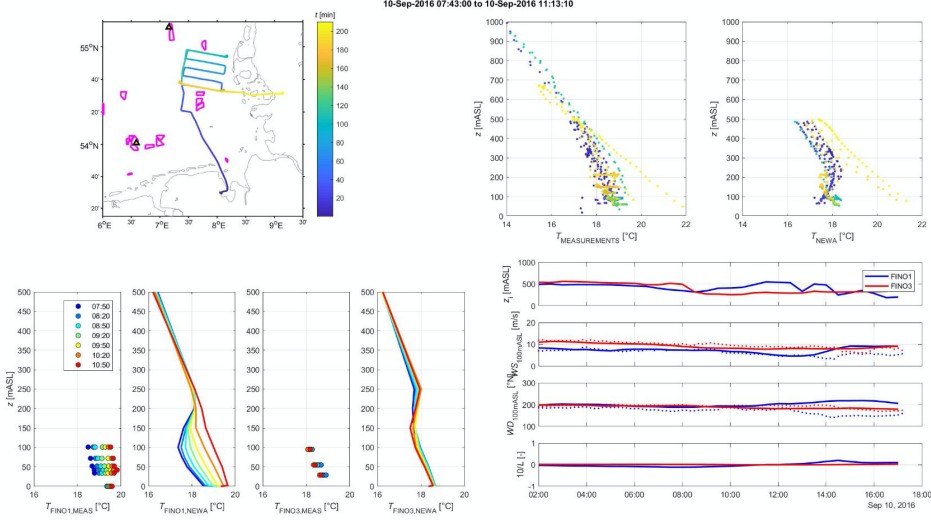


**Figure 7:** Comparison of temperature from in-situ measurements and NEWA model data over the German Bight from the
WIPAFF campaign on September 10, 2016. Clockwise from top-left: (a) Flight path with the location of existing wind farms
(indicated by magenta shapes) and the two met masts at FINO1 and FINO3 locations (black triangles in the southern and
northern part of the map, respectively). Line colour indicates time from flight start. (b) Comparison of temperature profiles
from in-situ measurements (flights) and NEWA model datasets (color-coded according to the corresponding flight time in
panel (a)). (c) Time series of ABL height, wind speed and direction at 100 m, and Obukhov length at surface provided by
ERA5 during the flight time period (blue line: FINO1; red line: FINO2). Wind speed and direction measured at 100 m from
met masts are also shown (dotted lines). (d) Comparison between temperature profiles from in-site measurements (met masts)
and NEWA model data during the flight time period (color-coded according to the corresponding flight time in panel (a)).





317

**3 Datasets and methodology**

**3.1 Microwave radiometer technology**

Microwave radiometry is a passive technique that has been used for several decades to observe atmospheric thermodynamic profiles. Ground-based microwave radiometers (MWR) are instruments measuring the down-welling natural thermal emission from the Earth's atmosphere, conveniently expressed in terms of brightness temperature ($T_B$), which is inverted into atmospheric thermodynamic products using statistical regression, neural network, or optimal estimation (Cimini et al., 2006). The ability to retrieve atmospheric variables depends upon the number and spectral allocation of the frequency channels at which the MWR measures $T_B$. The ability to retrieve atmospheric temperature profiles is related to thermal emission from oxygen, a well-stratified gas whose concentration is nearly constant in space, time, and height. Thus, radiation emitted by oxygen depends primarily on temperature, and $T_B$ measurements at channel frequencies exhibiting strong oxygen emission are highly correlated with atmospheric temperature. This is the case for the strong oxygen absorption complex at 50-70 GHz, which is well established and widely used for probing atmospheric temperature from the ground as well as from space. At channels in the centre of the absorption band (~60 GHz) the atmosphere is highly opaque and the observed $T_B$ carries information on the temperature near the instrument. Conversely, at channels away from the centre (e.g., 50-55 GHz), the atmosphere is less opaque and the signal systematically stems from atmospheric layers further from the instrument. Thus, vertical temperature profiles of the lower atmosphere are estimated from observations corresponding to different atmospheric absorption. The required information content can be obtained by multi-channel observations in the 50-60 GHz range but also by single-channel observations at several elevation angles. Similarly, observations at 22-32 GHz provide information on atmospheric humidity and column integrated water vapour (IWV) and liquid water path (LWP) simultaneously. Thus, ground-based MWR units operating in both the 22-32 GHz and 50-60 GHz bands are sometimes called MWR profilers (MWRP) and are commonly used to estimate atmospheric temperature and humidity profiles (Rüfenacht et al., 2021; Shrestha et al. 2021). A handful of MWR profiling types are currently available as off-the-shelf commercial products. Also a few research prototypes have been developed or are currently under development. For the scope of RAP, i.e. atmospheric profiling related to stability, only the temperature profilers and the MWRP are of interest. In our survey, we found only five commercially-available MWR products corresponding to these characteristics. These are listed in Table 1, together with their main characteristics. In addition, a prototype for marine deployment on a floating buoy or offshore platform is considered, though not commercially available yet.

**Table 1:** Main features of MWR types identified for potential interest for the atmospheric profiling related to stability (listed in alphabetical order of manufacturer). An estimate of the technology readiness level (TRL) is also shown. TLR 4-5 indicates





technology validated in the laboratory and relevant environment; TLR 9 indicates actual system proven in the operational
environment.

| Manufacturer | MWR name | Atmospheric retrievals | Range (km) | Type | TRL |
|---|---|---|---|---|---|
| Attex | MTP-5 | Temperature profile | <1 km | Single-channel; continuous elevation scanning. | 9 |
| BEST | MPR | Temperature profile IWV, LWP | <10 km | Multi-channel (2 polarisation); continuous elevation scanning. | 4-5 |
| Radiometrics | MP-2500A | Temperature profile | <10 km | Multi-channel; elevation scanning; optional azimuthal scanning. | 9 |
| Radiometrics | MP-3000A | Temperature profile Humidity profile IWV, LWP | <10 km | Multi-channel; elevation scanning; optional azimuthal scanning. | 9 |
| RPG | HATPRO | Temperature profile Humidity profile IWV, LWP | <10 km | Multi-channel; elevation scanning; optional azimuthal scanning. | 9 |
| RPG | TEMPRO | Temperature profile | <10 km | Multi-channel; elevation scanning; optional azimuthal scanning. | 9 |


For temperature profiles most of the information and the resolution resides in the first 2 km. Different methods
are used to quantify the vertical resolution of radiometric profiling. Using the inter-level covariance, Cimini et al.
(2006) reported that the vertical resolution of retrieved temperature profiles in the 0-3 km vertical range decreases
linearly with height $z$ as approximately $\sim 0.44 \cdot z$. Measurements at different elevation angles enhance the vertical
resolution of ABL temperature profile retrievals. Thus, elevation-angle scanning capability is often available in
MWRP units.
MWR units operate in all weather conditions. However, retrieved products may be unrealistic in case of water
accumulation over the radome, which produces additional microwave radiation not related with the atmospheric
state. A number of solutions for detecting and mitigating dew and precipitation effects are used in current MWR





instruments, including rain sensor, hydrophobic coating, tangent blower, heaters, shutter, and side-views. These
mitigation solutions effectively avoid water accumulation on the radome or mitigate its effect on the retrieved
products in most of the cases. However, chances are that mitigation solutions fail during intense rainfall or
snowfall. Proper maintenance (cleaning and replacing) of the radome helps in reducing cases of precipitation
mitigation failures. This requires regular services and replacement (e.g., every few months, depending upon
environment conditions). Off-shore conditions (high likelihood of sea sprays) may require more frequent
intervention.
A thorough assessment of MWR ability to provide atmospheric stability is given in Bianco et al. (2017),
specifically addressing wind energy applications. They report the outcome of a remote-sensing system evaluation
study, called XPIA (eXperimental Planetary boundary layer Instrument Assessment), held in spring 2015 at
NOAA's Boulder Atmospheric Observatory (BAO; Wolfe & Lataitis, 2018). BAO is equipped with a 300 m tower
mounting temperature and relative humidity sensors at six levels (50, 100, 150, 200, 250, and 300 m). In addition,
some 60 radiosondes were launched during the XPIA 2-month period. Two MWR of the same type (Radiometrics
MP3000-A, see Table 1) were deployed. To assess the MWR's ability to estimate atmospheric stability, they
compared MWR with tower measurements, analysing the vertical gradient of temperature T and potential
temperature $\theta$ for 50-300 m. For T gradient (dT/dz), they reported mean absolute error (MAE) within 2.1 K/km
and bias within 0.1 K/km, with 0.95 correlation. For potential temperature gradient (d$\theta$/dz), they reported MAE
within 2.2 K/km and bias within 0.1 K/km, with 0.95 correlation. They also investigated gradients for thinner
atmospheric layers (i.e., 50-150, 50-200, 50-250 m), reporting performances slightly degraded with respect to the
50-300 m layer. They also investigated the temperature profiling performances during rainy and non-rainy periods,
reporting no significant difference. They concluded that MWR can be useful for understanding conditions leading
to strong vertical windshear or turbulence, which can affect the loads on rotors. The next section extends the
results of Bianco et al. (2017) to other measurement conditions, including onshore and offshore.

### 386    3.2 Datasets

The results of Bianco et al. (2017) are obtained in a continental high-elevation site (Eire, Colorado, USA, ~1500
m altitude), using one of the MWR types in Table 1. This section aims to extend the analysis of Bianco et al.
(2017) to other environmental conditions and to the most common commercially available MWR system types in
Table 1. Thus, we identified datasets that would fit the purpose of validating MWR retrievals in different
environments, possibly both for onshore and offshore deployments. Several research and operational networks
operate onshore MWR continuously and provide open access to their data, e.g., the U.S. Atmospheric Radiation
Measurement (ARM, www.arm.gov) programme (Cadeddu et al., 2013), the European E-PROFILE programme
(Rüfenacht et., 2021), the New York State Mesonet (Shrestha et al., 2021). However, none of these MWR sites





are equipped with a 300 m tower as in BAO. Thus, the validation of MWR retrievals is here performed against in
situ measurements performed by balloon-borne radiosonde temperature sensors. Radiosondes are launched
routinely at a limited number of MWR sites and usually extend well above the altitude range relevant to wind
energy applications. Thus, we selected four datasets of colocated MWR and radiosonde observations taken at four
onshore sites including marine, continental, and Arctic environments: Graciosa island (Azores Archipelago,
Portugal), Saint-Symphorien (France), Lindenberg (Germany), and Pituffik (Greenland). Conversely, offshore
MWR deployments are rare, despite their potential for wind energy industry. To our knowledge, the only MWR
deployment on a fixed offshore platform was in the framework of the Offshore Boundary-Layer EXperiment at
FINO1 (OBLEX-F1, https://oblo.w.uib.no/activities/the-oblex-f1-measurment-campaign/), which took place
from May 2015 to September 2016 at the German wind energy research platform FINO1, in close vicinity to the
offshore wind park Alpha Ventus in the North Sea. The main purpose of the campaign was to improve
understanding of the marine boundary-layer in the vicinity of an offshore wind farm with respect to wind speed
profiles, atmospheric stability regimes, single turbine and wind farm wake propagation effects, under real offshore
conditions. To complement the resident instrumentation at the FINO1, several instruments were installed for the
campaign, including sonic anemometers, scanning wind lidars, and a MWR. The MWR (RPG HATPRO, see
Table 1) was deployed on the upper deck, at the base of the 100-m meteorological instrumented tower. However,
this dataset is not open access and the closest radiosondes are launched more than 50 km away from the coastal
site on the Norderney island (Germany). Conversely, colocated offshore MWR and radiosonde observations are
available from ship-based deployments, such as those performed in the framework of oceanic field experiments
(e.g., Bony et al., 2017). Thus, we selected two datasets of colocated MWR and radiosonde observations taken
from two research vessels (RV): the RV Polastern, going through the equator from northern Europe to southern
Africa or America in the framework of the OCEANET programme (Griesche et al., 2020), and the RV Meteor,
deployed offshore the Barbados in between the Caribbean sea and the Atlantic ocean (Schnitt et al., 2024) in the
framework of the EUREC[4]A (Elucidating the Role of Clouds-Circulation Coupling in Climate, Bony et al., 2017)
project. Other ship-based MWR deployments exist (e.g., Cimini et al., 2003; Yan et al., 2022) or are currently
being collected on a barge within the third Wind Forecast Improvement Project (WFIP3,
https://psl.noaa.gov/renewable_energy/wfip3/), but the datasets were not accessible to us at the time of this
analysis. More details about the considered datasets are given below, while Table 2 summarises the main
information. Note that the considered datasets include observations from three MWR types, covering all the MWR
manufacturers identified in Table 1.

**Table 2:** Main information on the datasets considered in this study.

| Dataset short name | Location | Environment | Deployment | Instruments | References |
|---|---|---|---|---|---|
| | | | | | |



| ENA | Graciosa Island, Azores (PT) | Marine, coastal, eastern north Atlantic | Onshore | MP3000-A | ARM, 2013; 2014 |
|---|---|---|---|---|---|
| MOL | Lindenberg (DE) | Continental, eastern Germany | Onshore | HATPRO | Güldner & Spänkuch, 2001 Vural et al., 2023 |
| SOF | St-Symphorien (FR) | Continental, south west France | Onshore | HATPRO MTP5 | Martinet et al., 2020 |
| PIT | Pituffik, Greenland (DK) | Arctic | Onshore | HATPRO | Pace et al., 2017 Pace et al., 2024 |
| POL | Polarstern RV | Open ocean, northern to southern Atlantic | Offshore | HATPRO | Griesche et al., 2020 |
| MET | Meteor RV | Open ocean, tropical | Offshore | HATPRO | Schnitt et al., 2024 Stephan et al., 2021 |


**ENA**: The Eastern North Atlantic (ENA) atmospheric observatory is located on Graciosa Island, part of the Azores
archipelago in the northeastern Atlantic Ocean west of Portugal. ENA is the newest atmospheric observatory
established by the U.S. ARM programme. The ENA observatory is a few hundred metres away from the coastline,
at 30 m altitude above mean sea level, and it is exposed to simil-ocean conditions throughout the year. The ENA
observatory also belongs to the Global Climate Observing System (GCOS) Reference Upper Air Network
(GRUAN), a network of several atmospheric observatories around the world providing reference-quality data for
climate benchmarking (Bodeker et al., 2015). ARM operates continuously a MWR (Radiometrics MP-3000 A,
see Table 1) and launches daily radiosondes from ENA (ARM, 2013; 2014). The dataset used here extends from
December 31st, 2018, to 15th March, 2019, for a total of 138 matchups between MWR and radiosonde
observations.

**MOL**: The Meteorological Observatory Lindenberg – Richard Aßmann Observatory (MOL-RAO) is operated by
the German Meteorological Service (Deutscher Wetterdienst, DWD). The MOL-RAO is located in the federal
state of Brandenburg in the north-eastern part of Germany, about 50 kilometres south-east of Berlin, 98 metres
above mean sea level. The MOL-RAO runs a comprehensive measurement program including all relevant surface
remote sensing and in-situ methods for studying solar and terrestrial radiation, interaction processes between the
Earth's surface and the atmosphere, and to produce the "Lindenberg Column", a reference dataset for
characterising the vertical structure of the atmosphere from the ground up to the stratosphere (e.g., Neisser et al.,
2002). The site contributes to all relevant national and international observational programs and initiatives such
as for instance the Aerosol, Clouds and Trace Gases Research Infrastructure (ACTRIS, Laj et al., 2024), Cloudnet





(Illingworth et al., 2007), the Baseline Surface Radiation Network (BSRN). MOL-RAO also hosts the lead center
of GRUAN, launching 4 radiosondes daily. The Lindenberg site provides a database of long-term MWR
observations of about 20 years (Güldner & Spänkuch, 2001) and operates currently two MWRs (Radiometrics
MP-3000A and RPG HATPRO G5, see Table 1). The dataset used here extends from September 1st, 2020, to 31st
December, 2020, for a total of 492 matchups between HATPRO MWR and radiosonde observations.

**SOF**: The SOuth west FOGs 3D experiment for processes study (SOFOG3D) is an international field campaign
directed by Méteo-France to advance our understanding of fog processes at the smallest scale to improve fog
forecasts by numerical weather prediction. SOFOG3D lasted from October 2019 to April 2020, during which an
unprecedented set of remote sensing and in-situ instruments was deployed during the whole winter period. A
unique network of eight MWR, was operated in a 300-by-300 km domain in the South-west of France (Martinet
et al., 2020; Martinet et al., 2022) for a better understanding of the spatio-temporal variability of fog at regional
scales and to conduct first data assimilation trials (Thomas et al., 2024). Two MWR were operated side-by-side
at the super-site, one HATPRO and one MTP5 (see Table 1). The dataset used here extends from 10 November
2019 to 12 March 2020, for a total of 61 matchups between two MWR units and radiosonde observations.

**PIT**: The Thule High Arctic Atmospheric Observatory (THAAO; https://www.thuleatmos-it.it/index.php) is
located within the U.S. Pituffik Space Base (formerly known as Thule Air Base) along the north-western coast of
Greenland (76,5°N, 68,8°W). The THAAO is on South Mountain, at 220 m above sea level and at about 3 and 11
km from the sea and from the Greenland ice sheet, respectively. THAAO is an international facility overseen by
the National Science Foundation which took over management in 2017 after the Danish Meteorological Institute
(DMI) discontinued their science activities at Pituffik. Research institutions from Italy (ENEA, INGV, University
of Roma "La Sapienza", University of Florence) and US (NCAR, AFRL) contribute to THAAO scientific
activities. The dataset used here was acquired in the frame of the SVAAP project (Study of the water VApour in
the polar AtmosPhere; Meloni et al. 2017) and extends from 12 July 2016 to 21 February 2017, for a total of 35
matchups between MWR and radiosonde observations.

**POL**: The ice breaker RV Polarstern is operated by the Alfred Wegener Institute for Polar and Marine Research
(AWI), and typically operates in the Arctic and Antarctic seas (Griesche et al., 2020; Engelmann et al., 2021;
Walbröl et al., 2022; and references therein). Atmospheric measurements are conducted en route to collect datasets
for investigating the energy budget between ocean and atmosphere and providing ground-truth information for
climate models. Continuous observations of aerosol, cloud, temperature and humidity profiles, liquid-water path,
solar and thermal radiation, sensible and latent heat are performed. The remote-sensing instruments are hosted in
a sea container deployed at the upper deck, starboard of Polarstern at about 22 m above sea level, called the





OCEANET platform. OCEANET houses an extensive suite of ground-based remote-sensing instruments,
including a multiwavelength Raman polarisation lidar and one 14-channel microwave radiometer (RPG
HATPRO, see Table 1). Polarstern also hosts a SCalable Automatic Weather Station (SCAWS), belonging to
DWD, which includes a radiosonde launching system. One radiosonde per day is launched routinely from the
deck of the Polarstern RV, between 11-12 UTC, but additional launches are occasionally performed earlier or
later in the day (e.g., ~09 or 22 UTC). The considered cruises swept the Atlantic Ocean from north to south and
return. The dataset used here were collected during sixteen 2-month cruise missions, extending from 20 April
2007 to 9 December 2016, for a total of 316 matchups between MWR and radiosonde observations.

**MET**: The RV Meteor participated in the EUREC[4]A project (Bony et al., 2017; Stevens et al., 2021), a 5-week
campaign in the Tropical Atlantic windward and in the close vicinity of Barbados, which included ship-based
MWR (Schnitt et al., 2024) and radiosonde (Stephan et al., 2021) observations. During EUREC[4]A (January to
February, 2020), MWR measurements aboard the RV Meteor were performed by a HATPRO G5 operated by the
Leipzig Institute for Meteorology, so called LIMHAT. The LIMHAT MWR was placed on the navigation deck
of the ship at 15.8 m above sea level, operated at a temporal resolution of 1s in zenith mode, with elevation scans
performed every full hour. Radiosondes were also launched from the same deck. Before February 9[th], radiosondes
were launched from the port side of the ship, and after that date, from the stern of the ship due to the failure of the
sonde container (Stephan et al, 2021). A linear regression was used to retrieve temperature profiles (Schnitt et al.,
2024; Walbröl et al., 2022), trained with a large dataset of daily radiosoundings launched from 1990 until 2018
from Grantley Adams International Airport in Barbados (station ID 78954 TBPB). The dataset used here extends
from 16 January to 1 March 2020, including 219 radiosondes, providing a total of 145 (68) matchups between
radiosonde observations and MWR zenith (elevation scan) retrievals.

**3.3 Methodology**
Following Bianco et al. (2017), the MWR ability to provide atmospheric stability is assessed through the analysis
of vertical gradients of atmospheric temperature (dT/dz) and potential temperature (d$\theta$/dz) in the 50-300 m vertical
range. Here, the potential temperature profile is calculated using Eq.(2) with $P_0$=1000 mb and $R/c_p$=0.286. The
profiles of $T$ (in K) and $P$ (in mb) are given by the temperature profile retrieved from the MWR and the pressure
profile estimated via the atmospheric thickness equation (with the temperature retrievals and the surface pressure
measured by the sensor embedded within the MWR as inputs). For all the datasets we consider radiosondes as
reference measurements for atmospheric temperature and potential temperature. Potential temperature from
radiosondes is computed as above but using temperature and pressure measurements from the radiosonde sensors.
Temporal colocation between MWR measurements and radiosonde data is achieved averaging the MWR



measurements within 30 minutes after the radiosonde launch. For spatial colocation, radiosonde data are
interpolated on the vertical grid defined for MWR profile retrievals. Examples of simultaneous MWR and
radiosonde profiles for temperature and potential temperature are shown in Figure 8, for two of the considered
datasets (ENA and SOF) including the three most common commercially-available MWR types. Figure 8
indicates that MWR can generally reproduce the structure of both temperature and potential temperature profiles,
although at a lower vertical resolution. Looking at the potential temperature profiles, the two selected cases
correspond to classic unstable and neutral/stable atmospheric conditions (Stull, 2012). For each of the available
datasets, we produce couplets of T and $\theta$ profiles from MWR and radiosonde, from which statistical agreement is
computed in terms of vertical profiles of bias, standard deviation (STD), and RMS difference. For each couplet,
vertical gradients between 50-300 m are computed (dT/dz and d$\theta$/dz) from both MWR and radiosonde profiles.
Figure 9 shows a 2.5-month time series of d$\theta$/dz at ENA site as computed from MWR and radiosondes. The
statistical agreement is then computed in terms of mean average (AVG), STD, RMS and maximum absolute error
(MAE). Typical uncertainty of radiosonde temperature measurements below 5 km is ~0.2-0.5 K (Dirksen et al.,
2014). Thus, assuming uncorrelated uncertainty at different layers, the uncertainty of temperature gradients from
radiosonde is estimated as ~1.1-2.8 K/km. However, the representativeness uncertainty, resulting from the
representation of an air volume with radiosonde point measurements, is probably dominating and more difficult
to estimate generically, as it depends on site climatology and meteorological conditions.

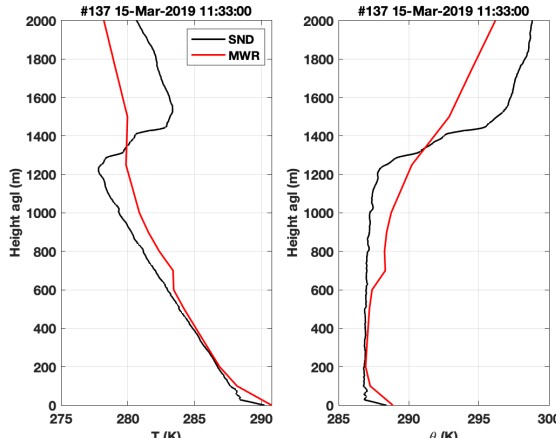




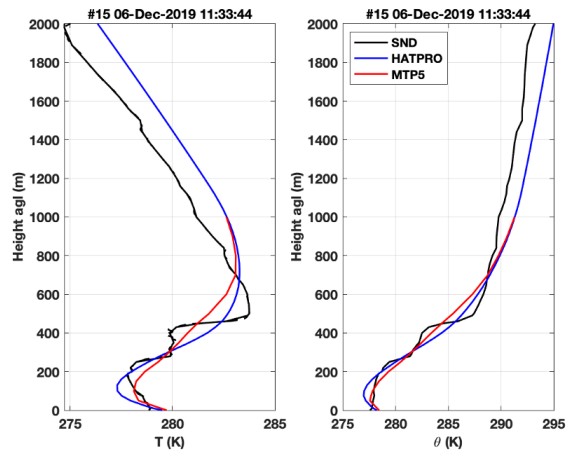


**Figure 8:** Simultaneous temperature (left) and potential temperature (right) profiles from radiosonde (black) and three MWR

types. Top: MP3000-A (red) at the ENA site (unstable conditions). Bottom: HATPRO (blue) and MTP5 (red) at the

SOFOG3D supersite (neutral to stable conditions). Note that MTP5 retrievals are limited to 1-km height.


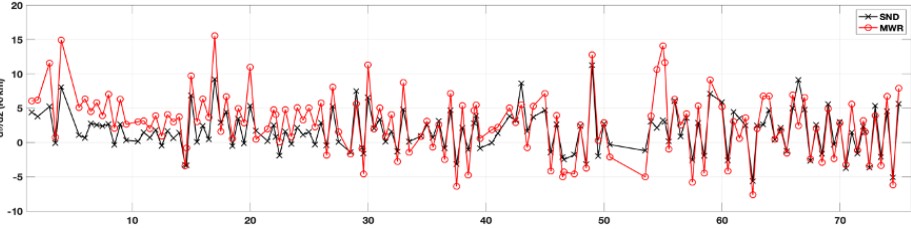


**Figure 9:** 2.5-month time series of potential temperature lapse rate ($d\theta/dz$) between 0 and 300 m a.g.l. derived from MWR

temperature retrievals (red line) and from radiosonde observations (black line). Dataset from Graciosa Island from 1 January

to 15 March, 2019.


**4 Validation**

This Section presents quantitative results of the statistical analysis on the ability of MWR to provide atmospheric

temperature and potential temperature profiles and vertical gradients, which are related to the atmospheric

stability. The results are discussed below separately for each dataset.




**ENA:** The first considered dataset was collected at the ENA observatory, located a few hundred metres away
from the northern coastline of Graciosa Island in the Eastern North Atlantic, conveniently exposed to Atlantic
ocean conditions throughout the year. The considered dataset of MWR profiler and radiosonde observations spans
about 3 months (from 2019-01-01 to 2019-03-15). The MWR is a Radiometrics MP3000-A (see Table 1). Two
radiosondes per day are launched from ENA at ~11:30 and 23:30 UTC, providing 138 matchups between MWR
retrievals and radiosonde profiles in the considered period. From the set of 138 matchups, statistics for temperature
and potential temperature profile accuracy are calculated. Accordingly, for the ENA dataset Figure 10 reports the
vertical profiles of bias, STD, and RMS difference between temperature and potential temperature profiles
measured by radiosondes and estimated by MWR. The scores for temperature profile retrievals are in line with
those available from the open literature (Cimini et al., 2006; Löhnert and Maier, 2012; Bianco et al., 2017). The
scores for potential temperature profiles are very similar to those for temperature profiles, though not exactly the
same due to the influence of pressure profile (measured by radiosondes while estimated from surface pressure and
retrieved temperature by MWR). Figure 11 reports the scatter plot of temperature gradient (dT/dz) and potential
temperature gradient (d$\theta$/dz) in the vertical range (50-300 m). It shows that MWR estimates of either dT/dz or
d$\theta$/dz are correlated with radiosonde measurements throughout the spanned range, with larger scatter towards
higher values. The range of d$\theta$/dz goes from negative to positive values (indicatively from -5 to +15 K/km), i.e.
from atmospheric stable through neutral to unstable conditions. The statistical results are computed from the two
samples of dT/dz and d$\theta$/dz couplets in terms of AVG, STD, RMS, and MAE. A summary from all the considered
datasets is reported in Table 3. For convenience, Table 3 also reports the statistical results from Bianco et al.
(2017), as obtained from the XPIA dataset from Colorado (USA). For the ENA datasets, these can be summarised
as follows: for both temperature gradient (dT/dz) and potential temperature gradient (d$\theta$/dz), the MAE results
within 2.4 K/km, bias within -1.2 K/km, with 0.72 correlation. These performances are somewhat worse than
those reported by Bianco et al. (2017) for XPIA, i.e. MAE within 2.2 K/km, bias within -0.1 K/km, with 0.95
correlation. Note that the same MWR type operates at the two sites (MP-3000A), but the notable difference may
be related to the status of the instrument calibration and/or the appropriate fitting of the retrieval coefficients to
the different climatology conditions (ENA: winter marine environment; XPIA: spring mountain environment).



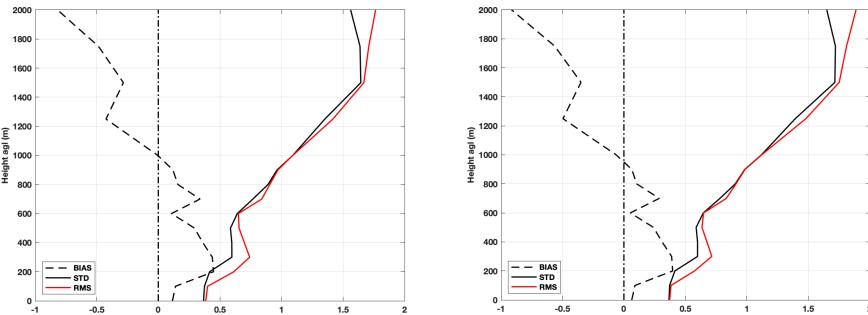


**Figure 10:** (Left) Bias, standard deviation (STD), and root-mean-square (RMS) differences of the MWR-minus-radiosonde

temperature residuals from the 138 matchups collected at the ENA observatory on Graciosa Island (Eastern North Atlantic)

from 2019-01-01 to 2019-03-15. (Right) Same but for potential temperature profiles.


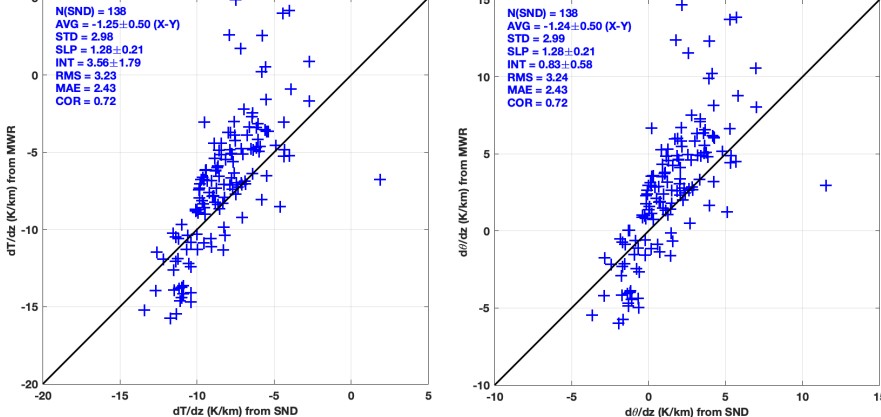


**Figure 11:** Comparison of atmospheric lapse (50–300 m) for temperature (left) and potential temperature (right) for

MWR retrievals vs. radiosonde measurements collected at the ENA observatory on Graciosa Island from 2019-01-01 to

2019-03-15. Text within each panel as in Figure 5. Units for AVG, STD, RMS, and MAE are in K/km.


**MOL:** This dataset was collected at the MOL in north-eastern Germany, about 98 metres above mean sea level,

characterised by typical mid-latitude continental climatology conditions. The considered dataset of MWR profiler

and radiosonde observations spans about 4 months (from 2020-09-01 to 2020-12-31). The MWR is a RPG Hatpro



G5 (see Table 1). Four radiosondes per day are launched at ~5:30, 11:30, 17:30 and 23:30 UTC, providing 492
matchups between MWR retrievals and radiosonde profiles. From the set of 492 matchups, statistics for
temperature profile accuracy are calculated and reported in Figure 12a, similarly to Figure 10. Also for this dataset,
the scores for temperature profile retrievals are in line with those available from the open literature, though the
STD/RMS increases more rapidly in the 200-1400 m vertical range. The statistics for the potential temperature
profiles are almost identical to those for temperature and thus are not shown for this nor for the remaining datasets.
Scatter plots of $dT/dz$ and $d\theta/dz$ from MWR and radiosondes are reported in Figure 13a. As for ENA, the MOL
dataset corresponds to different climatology (autumn continental environment) with respect to that of XPIA. The
behaviour of both $dT/dz$ and $d\theta/dz$ are similar for the ENA and MOL sites, though showing higher correlation at
MOL (0.91) than at ENA (0.72).

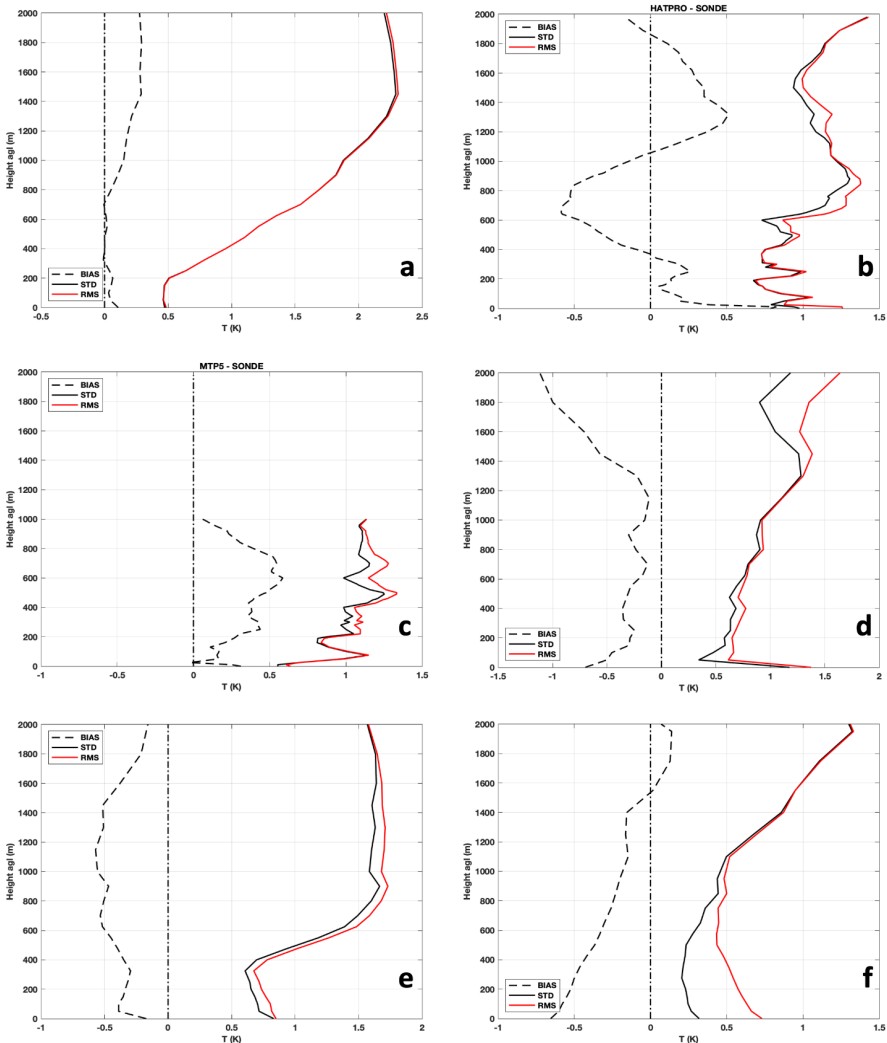

**Figure 12:** Profiling performances for temperature profiles as in Fig. 8, but obtained from the other considered datasets: (a) 492 matchups collected at MOL (Lindenberg, Germany) from 2020-09-01 to 2020-12-31. (b) 61 matchups during the SOFOG3D campaign ( Saint-Symphorien, France, October 2019 to April 2020) for the HATPRO MWR. (c) 61 matchups during the SOFOG3D campaign, but for the MTP-5 MWR (limited to 1 km altitude above ground). (d) 35 matchups during the SVAAP project (2016-07-12 to 2017-02-21) collected at Pituffik (Greenland). (e) 298 matchups from sixteen Polarstern





RV cruises from 2007 to 2016. (f) 145 matchups from the RV Meteor during the EUREC[4]A campaign (from 2020-01-16 to
2020-03-01, zenith-mode only).

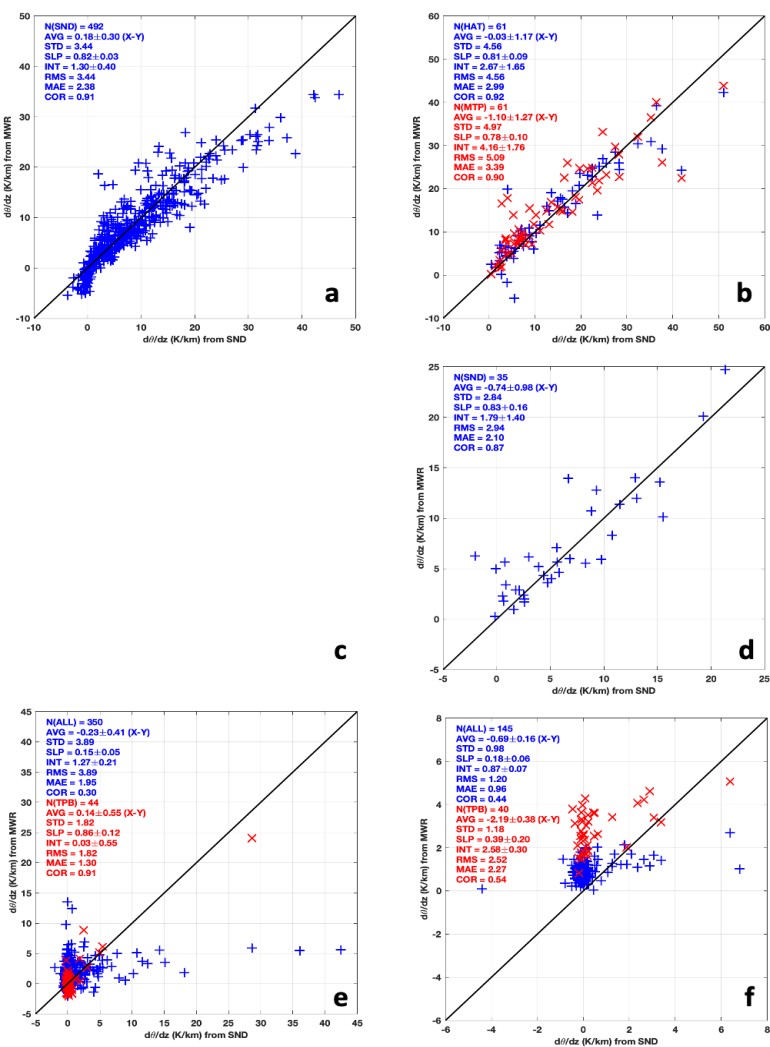


**Figure 13:** Comparison of atmospheric potential temperature lapse rate as in Fig.9 but for MWR retrievals vs. radiosonde
measurements collected at other sites: (a) MOL (Lindenberg, Germany). (b) SOFOG3D campaign (Saint-Symphorien,


France); blue crosses indicate HATPRO data, red Xs indicate MTP-5 data. (c) This panel is left intentionally blank. (d)
SVAAP project (Pituffik, Greenland). (e) Polarstern RV cruises (North-South Atlantic); blue crosses indicate all retrieval at
zenith, red Xs indicate elevation scan retrievals (2016 only). (f) Meteor RV during EUREC[4]A campaign (Barbados); blue
crosses indicate retrievals at zenith, red Xs indicate elevation scan retrievals. Text within each panel as in Figure 5. Units for
AVG, STD, RMS, and MAE are in K/km.
**SOF:** The same analysis is performed on the dataset collected during the Météo-France SOFOG3D international
field campaign in South-west of France. Two MWR were operated side-by-side at the supersite in Saint-
Symphorien, one HATPRO and one MTP5 (see Table 1). 61 radiosondes were launched, mostly during stable
conditions prone to fog formation during the period from 2019-11-10 to 2020-03-10. Statistical comparison of the
61 radiosonde profiles with nearly simultaneous MWR retrievals from both the HATPRO and MTP5 are reported
in Figure 12b-c. Note that retrievals from MTP5 are limited to 1 km altitude, while retrievals from HATPRO are
provided up to 10 km (although the sensitivity drops to negligible values above 2-3 km). For the vertical range
covered by both radiometers (< 1 km), their performances are quite similar (in terms of bias, STD, and RMS),
with slightly better performances close to the surface for the MTP5. Statistics for temperature and potential
temperature gradients in the 50-300 m vertical range during the SOFOG3D experiment are shown in Figure 13b.
As for the temperature profiles, also for the gradients the performances of the two radiometers are quite similar.
The HATPRO shows slightly higher scores (e.g, ~2% increase in correlation) than MTP5, despite the slightly
better profiling performances of the MTP5 near the surface.
**PIT:** This dataset was collected at the THAAO within the U.S. Pituffik space base along the north-western coast
of Greenland, at 220 m above sea level, characterised by typical Arctic climatology conditions. The MWR is a
RPG Hatpro G2 (see Table 1). During the SVAAP project (2016-07-12 to 2017-02-21), radiosondes were
launched sporadically during clear-sky conditions, with a total of 35 matchups between MWR retrievals and
radiosonde observations. Statistics for temperature and profile accuracy are calculated and reported in Figure 12d.
Also in this case, the scores for temperature profile retrievals are in line with those available from the open
literature, though slightly larger than expected near the surface. Figure 13d reports the scatter plot of potential
temperature gradient (d$\theta$/dz). This dataset corresponds to yet another climatology (polar environment) with
respect to the previous ones. The statistical scores for both dT/dz and d$\theta$/dz are similar to the previous sites, higher
than ENA but slightly lower than MOL/SOF in terms of correlation (~0.87).
**POL:** This dataset consists of MWR and radiosonde data from sixteen Polarstern RV cruises (from 2007 to 2016)
from northern to southern Atlantic, across the Equator. One radiosonde per day was launched routinely between
11-12 UTC, but other launches were performed occasionally. A total of 466 radiosonde launches have been
collected during the sixteen cruises, leading to 365 matchups with MWR data, of which 350 survived a quality



control screening. From the set of 350 matchups, statistics for temperature profile accuracy are calculated and
reported in Figure 12e. The statistics for temperature profile retrievals are larger than those available from the
open literature, especially below 500 m. While the systematic component (bias) stays within 0.5 K, the random
component (STD) presents a peak near to the surface, leading to ~0.8 K RMS. This feature naturally affects the
comparison of temperature and potential temperature gradients. Figure 13e reports the scatter plot of d$\theta$/dz
measured by the MWR and the radiosondes, clearly showing low correlation (~0.3). It appears that except for few
cases, the radiosondes measure nearly neutral stability (i.e., d$\theta$/dz~0 K/km) while the MWR reports all the range
from slightly unstable (d$\theta$/dz<0 K/km) to very stable conditions (d$\theta$/dz>0 K/km). In addition, for the few cases
in which radiosondes measure very stable conditions (d$\theta$/dz>10 K/km), the MWR retrievals seem to saturate at
~5 K/km. One possible cause may be the zenith-only observation mode adopted during these Polarstern RV
cruises. In fact, although elevation scanning observations are proved to increase the accuracy of MWR temperature
retrievals (Cimini et al., 2006), especially below 1 km, the zenith-only mode was chosen aboard the Polarstern
RV to avoid mispointing problems caused by the ship pitch and roll movements. This cause can be investigated
by analysing further the dataset of Polastern RV data collected during the two cruises in 2016, when elevation
scanning observations were also available. The analysis of this additional dataset, corresponding to MWR
retrievals from elevation scanning observations during the two cruises of 2016, is also reported in Figure 13e.
Although the scatter of potential temperature gradients seems similar, the statistical scores of elevation scanning
retrievals improve substantially with respect to zenith only, in terms of RMS (from 3.78 to 1.84 K/km), MAE
(from 1.97 to 1.30 K/km), and correlation (from 0.31 to 0.90), though the latter is mostly driven by only one point
(at 27 K/km). Although limited, this dataset seems to confirm that elevation scanning is indeed desirable for off-
shore MWR deployment. Another possible cause of the rather poor performances may be related to the dataset
used to train the inversion method (multiple regression). As detailed in Doktorowski (2017), the training is based
on a homogenised dataset of 2621 radiosondes launched from cargo vessels in all climatic zones between 60N
and 60S, which may be too broad to represent the peculiar environmental conditions encountered by the Polarstern
during the sixteen cruises from 2007 to 2016. In particular, the training set may under-represent the deep neutral
conditions which seem to characterise most of the radiosonde profiles during the Polarstern RV cruises.
**MET:** Another ship-based dataset of colocated MWR and radiosonde observations is available from the RV
Meteor during the EUREC[4]A project. 219 radiosondes were launched from the RV Meteor between 2020-01-16
and 2020-03-01, corresponding to typical tropical conditions. The LIMHAT Level 3 version 2.0 dataset is used
here (Schnitt et al, 2023). From this dataset, 145 matchups between radiosonde observations and MWR zenith
temperature profile retrievals are available, for which the statistical agreement is calculated and reported in Figure
12f. STD for temperature profile retrievals is in line with the expectations from the open literature, while the bias
presents a ~0.7 K peak near to the surface, dominating the RMS in the lower 500 m. The scatter plot of potential





temperature gradients is reported in Figure 13f, for both the zenith-mode (145 matchups) and elevation-mode
retrievals (40 matchups). Similarly to POL, radiosonde data indicate dominant nearly-neutral conditions (d$\theta$/dz~0
K/km), while MWR data mostly indicate slightly stable conditions (d$\theta$/dz~0-4 K/km). For the few cases where
radiosondes indicate either unstable (d$\theta$/dz~-4 K/km) or stable conditions (d$\theta$/dz~7 K/km), the zenith-mode data
remain with 0-3 K/km, resulting in low correlation overall (0.44). Correlation is slightly larger for elevation-mode
retrievals (0.54), but also MAE is larger (2.27 K/km) due to a ~3-time larger AVG. Note that, while theory and
previous field campaigns have shown that elevation scans should improve the retrieved temperature profiles in
the lowest kilometre (Cimini et al. 2006), this is the opposite for the EUREC[4]A LIMHAT dataset. In fact, as
reported by Schnitt et al. 2023, bias and RMS for the elevation-mode retrievals increase substantially with respect
to zenith-mode (by a factor of 2 near the surface, see their Fig. 9). The authors attribute this to the training set
(radiosondes launched from Grantley Adams International Airport), which may be impacted by an island effect,
leading to warmer temperatures near the surface compared to the zenith column over the ocean. Another potential
reason is the ship pitch and roll movements, since the LIMHAT was not stabilised, which may especially affect
observations at low elevation angles.

**Table 3:** Summary of the statistics for temperature and potential temperature gradients from MWR validated against
radiosonde measurements (50-300 m AGL). Note that for XPIA, the correlation coefficient is derived from the coefficient of
determination ($R^2$) given in Bianco et al., 2017. POL(ZNT-ALL) indicates zenith-only MWR retrievals from all 16 Polastern
cruises (2007-2016), while POL(ELV-2016) indicates elevation-scan MWR retrievals from two Polastern cruises in 2016.

| | Temperature gradients | | | Potential temperature gradients | | |
|---|---|---|---|---|---|---|
| **Dataset** | **Bias (K/km)** | **MAE (K/km)** | **Correlation** | **Bias (K/km)** | **MAE (K/km)** | **Correlation** |
| **XPIA** | 0.10 | 2.10 | 0.95 | 0.10 | 2.20 | 0.95 |
| **ENA** | -1.25 | 2.43 | 0.72 | -1.24 | 2.43 | 0.72 |
| **MOL** | 0.16 | 2.36 | 0.91 | 0.18 | 2.38 | 0.91 |
| **SOF (HATPRO)** | 0.00 | 2.97 | 0.92 | -0.01 | 2.99 | 0.92 |
| **SOF (MTP5)** | -1.06 | 3.37 | 0.90 | -1.10 | 3.39 | 0.90 |
| **PIT** | -0.75 | 2.06 | 0.88 | -0.74 | 2.10 | 0.87 |
| **POL(ZNT-ALL)** | -0.21 | 1.93 | 0.30 | -0.23 | 1.95 | 0.30 |
| **POL(ELV-2016)** | 0.32 | 1.42 | 0.89 | 0.14 | 1.30 | 0.91 |
| **MET(ZNT)** | -0.59 | 0.88 | 0.44 | -0.69 | 0.96 | 0.44 |





| | | | | | | |
|---|---|---|---|---|---|---|
| **MET(ELV)** | -2.10 | 2.19 | 0.54 | -2.19 | 2.27 | 0.54 |


Finally, the scores for temperature and potential temperature gradients from all the datasets, including the
reference from Bianco et al. (2017), are reported in Table 3. Note that the range of temperature gradients is quite
different in the seven datasets (~30 K/km for XPIA, ~20 K/km for ENA, ~50 K/km for MOL and SOF, ~25 K/km
for PIT, ~40 K/km for POL, and ~14 K/km for MET), which affects the values of RMS, MAE, and correlation.
The statistics from MOL and SOF (continental mid-latitude sites, winter to spring) are similar, and just slightly
lower than those reported for XPIA (mountain site, spring). For the onshore datasets (top six rows in Table 3),
potential temperature gradients agree with those from radiosondes with correlation ranging from 0.72 to 0.95 and
MAE from 2.10 to 3.39 K/km. The lowest correlation (0.72) corresponds to ENA (winter marine environment),
while for all the others correlation is higher than 0.88. This gives some confidence that MWR performances are
site independent, provided that the radiometer and inversion method are properly calibrated and trained,
respectively. Conversely, the ship-borne datasets (bottom four rows in Table 3) provide substantially lower
correlation considering zenith retrievals (0.30 to 0.44). Elevation scanning seems beneficial, increasing correlation
(from 0.3 to 0.9 for POL, 0.4 to 0.5 for MET), though for POL is mostly driven by one matchup only and for MET
it comes at the expense of ~2-time larger MAE. Note that the MWR retrieval algorithm for the POL and MET
datasets is the same (linear regression), though trained with independent datasets (POL: 2621 ship-borne
radiosondes; MET: 10871 radiosondes launched from an island-based airport). This suggests that appropriate
dedicated training and elevation scanning with ship movement compensation may be required for MWR to catch
potential temperature gradients typical of off-shore conditions.

**5 Summary, conclusions, and outlook**

Atmospheric stability is relevant for wind energy applications, as it influences the propagation of wind turbine
wakes. Wind turbine rotors operate in the lowest 300 m, and atmospheric stability below and above that height
may influence their operations through vertical wind shear and turbulence. Considering different power curves
for different stability conditions leads to more accurate and reliable performances of energy production, which
lowers the financial risks for both operators and manufacturers. Thus, the ability to model and measure
atmospheric stability was reviewed using available datasets of reanalysis and mesoscale NWP model output, tower
measurements, and ground-based remote sensing observations.
Surface stability metrics from model datasets, including NWP (NEWA and DOWA) and global reanalysis
(ERA5), have been assessed against measurements from met masts and floating lidar, focusing on the Obukhov
length. The results confirm that when the main drivers of atmospheric stability are correctly characterised by the



bulk formulations used in NWP models, the modelled Obukhov length time series compare to those derived from
measurements. Overall, the best match between model data and measurements is observed for ERA5 datasets, in
particular computed from the fluxes for unstable conditions and using the bulk Richardson number for stable
conditions. Two examples are reported to illustrate how the modelled Obukhov length time series can improve
wind-related analyses. The first demonstrates how the atmospheric stability class indicated by the modelled
Obukhov length correlates to turbulence intensity and wind speed spectra, both progressively increasing as
conditions shift from stable to neutral to unstable. The second example shows that surface-layer expressions, such
as Monin-Obukhov Similarity Theory, predict reasonably the wind speed profile in neutral and unstable
conditions, while significantly overpredict wind speed measurements in stable conditions, requiring additional
information on upper air effects (e.g. the boundary layer height) to better capture the wind speed above 30 m. The
ability of NWP models to characterise air temperature profiles in different stability conditions was assessed in the
30-100 m vertical range against tower measurements (at FINO1/FINO2 platforms). Both DOWA and NEWA are
quite accurate on average, with mean differences of ~0.3-0.5 K with respect to measurements, with no clear pattern
with respect to the stability class. Conversely, both DOWA and NEWA models show increased RMS in stable
conditions with respect to unstable conditions, with a minimum RMS in neutral conditions. DOWA performs
better than NEWA, the first showing RMS within 1 K regardless of stability conditions, while the second showing
RMS up to 2.2 K, especially in very stable conditions. Also for temperature gradients in the 50-100 m layer, the
DOWA performs better than NEWA, as measured by MAE (3.4-4.0 K/km for DOWA, 3.5-4.2 K/km for NEWA),
RMS (5.8-7.3 K/km for DOWA, 6.4-8.4 K/km for NEWA), and correlation (0.77-0.80 for DOWA, 0.70-0.71 for
NEWA).
Thus, it is concluded that reanalysis and NWP models do provide wind energy practitioners with useful
information on atmospheric stability (e.g., Obukhov length) for many situations, i.e., the mean can be used for a
range of analyses, including estimates of turbulence and wind shear. However, in specific cases (e.g., elevated
temperature inversion) and especially during near-surface stable stratification, the simulated profiles may not be
sufficiently accurate. Typical conditions for difficult stability characterization have been illustrated using datasets
of surface wind from SAR observations and in situ temperature/wind profiles from UAS measurements. Cases
with long wind farm wakes, as they typically occur in a stably stratified ABL, have been identified when
observations and models at surface indicate unstable and neutral conditions, suggesting the need for continuous
measurements above the height of typical met mast (~100m).
This need can be satisfied by nearly continuous observations from ground-based remote sensing atmospheric
profilers, and this study addresses the specific question: How good are atmospheric stability retrievals from
microwave radiometer measurements for wind energy applications in different climates? Here, the ability of
commercially-available MWR to profile atmospheric temperature within the first 2 km and to provide potential





temperature gradients in the vertical range of wind turbine rotors has been assessed against in situ radiosonde
measurements. Several sources of MWR data have been identified and analysed, giving preference to datasets in
different environments and climatological conditions and datasets with observations from all identified MWR
manufacturers. This analysis extends the results in Bianco et al. (2017), obtained for the MP3000A deployed in a
continental high-elevation site (~1500 m, Colorado, USA), to other MWR types and environmental conditions. In
total, six datasets are considered here, of which four are for onshore and two for offshore environments. The four
onshore include marine (east-northern Atlantic), continental (north-eastern Germany; south-west France), and
Arctic (Greenland) environments. The two offshore datasets are collected from two research vessels: the Polastern,
cruising the Atlantic from northern Europe to southern Africa/America, and the Meteor, deployed off the coast of
Barbados in the Caribbean sea. The considered datasets include observations from all the identified commercial
MWR types (i.e., HATPRO, MP3000A, MTP5). From the analysis of the six datasets considered in this study, we
conclude that:

1)  The statistics for temperature profile retrievals are mostly in line with those available from the open
literature, i.e., bias within +/-0.5 K and RMS ~0.5 K near the surface increasing to ~1.5 K at 2 km,
although with some exceptions (e.g., higher bias and RMS near the surface for HATPRO in SOF and
PIF). Statistics from NWP models in the 30-100m altitude range show similar biases but larger RMS
(increasingly larger than 0.5 K from unstable to stable conditions, especially for NEWA).
2)  For the onshore datasets, potential temperature gradients agree with those from radiosondes with
correlation ranging from 0.7 to 0.9 and MAE from 2.1 to 3.4 K/km. This mostly confirms the results of
a previous study (Bianco et al., 2017), limited to one onshore dataset and one MWR type. Similar
performances from sites in different environments and with different climatology give some confidence
that MWR performances can be considered site independent, provided that the radiometer and inversion
method are properly calibrated and trained, respectively.
3)  For the offshore datasets, considering zenith retrievals the MAE is relatively small (0.9 to 1.9 K/km)
while the correlation is substantially lower (0.3 to 0.4). The low performances are partially due to the
relatively narrow range of potential temperature gradients from radiosondes, indicating prevailing neutral
conditions. This poses a question on the datasets used to train the inversion algorithm, as global or
onshore datasets may under-represent the prevailing neutral conditions shown by the offshore datasets
available here.
4)  Again for the offshore datasets, elevation scanning seems beneficial, increasing correlation (from 0.3 to
0.9 for POL, 0.4 to 0.5 for MET). For POL, elevation scanning also decreases MAE, while for MET
MAE increases by a factor ~2, due to a 3-time larger AVE. This may also be related to the training data

set, which could be affected by an island effect, but also to the ship movement (pitch and roll), which

may have some impact on low-elevation observations.

5)   Considering all the six datasets, the MAE between MWR and radiosonde temperature (and potential

temperature) gradients in the 50-300 m vertical range goes from 0.9 to 3.4 K/km, while the RMS

difference from 1.2 to 5.1 K/km. The latter includes the uncertainty of the radiosonde temperature sensor

(1.1-2.8 K/km). Considering this, the uncertainty of MWR for temperature and potential temperature

gradients in the 50-300 m vertical range is estimated between ~0.5-4.3 K/km.

This study indicated the lack of systematic off-shore MWR measurements. Systematic off-shore MWR
measurements are needed to enlarge the range of meteorological conditions and to characterise the performances
under different stability stratifications. The conclusions above indicate that appropriate dedicated training and
elevation scanning (with movement compensation, if ship-based) may be required for MWR to catch potential
temperature gradients typical of off-shore conditions. Wind energy practitioners may be interested in learning
what instrument is best when and where. To address this properly, we would need to have the different MWR
types running at the same time in different environments with the same retrieval method. To our knowledge, no
such a dataset is currently available, nor plans to implement such an intercomparison. However, other onshore
and offshore MWR observation datasets may be exploited to extend this analysis, characterising performances in
other conditions and testing optimization strategies, e.g., in the context of the MiradOR (microwave radiometers
for assessing offshore wind resources) project, currently under evaluation. Also, instrument synergy may be
exploited to increase vertical resolution of temperature profiles and thus improve retrieval performances of
temperature gradients, as shown onshore for combined passive (MWR and IRS) and active (RASS) sensors
(Turner and Löhnert, 2021; Bianco et al., 2024), although not all these instruments are practical to be deployed
offshore. From the above perspectives, one of the most valuable datasets up to date is the one produced recently
by the 3rd Wind Forecast Improvement Project (https://psl.noaa.gov/renewable_energy/wfip3/), including MWR,
IRS, and several active instruments deployed over a barge off the coast of southern New England.

**Competing interests**
Some authors are members of the editorial board of AMT. The research was funded by the Carbon Trust as part
of the Offshore Wind Accelerator (OWA) program, supported by the following partner companies (in alphabetical
order): EnBW, Equinor, Orsted, RWE, Scottish Power Renewables, Shell, SSE Renewables, Total Energies,
Vattenfall.





## Author contribution

Conceptualization and funding acquisition: DC, RG, and SF acquired the funding, designed and lead the research. Data curation: CA, AB, CK, PM, GP, BP provided experimental data and performed data curation. Visualization: DC, RG, and AB created the figures. Supervision and validation: SG, EG, STN, and FR oversaw the research activity planning and execution, including mentorship external to the core team. Writing: DC, RG, and SF prepared the manuscript original draft, which was reviewed and edited by all co-authors.

## Acknowledgements

This research was funded by the Carbon Trust as part of the Offshore Wind Accelerator (OWA) Radiometry and Atmospheric Profiling (RAP) project. OWA partner companies are acknowledged: (in alphabetical order) EnBW, Equinor, Orsted, RWE, Scottish Power Renewables, Shell, SSE Renewables, Total Energies, Vattenfall. The research was stimulated by COST Action CA18235 PROBE (https://probe-cost.eu/), supported by COST (European Cooperation in Science and Technology, www.cost.eu). Data at ENA were obtained from the Atmospheric Radiation Measurement (ARM) user facility, a U.S. Department of Energy (DOE) Office of Science user facility managed by the Biological and Environmental Research Program. The SOFOG3D field campaign was supported by METEO-FRANCE and ANR through grant AAPG 2018-CE01-0004. The MWR network deployment during SOFOG3D was carried out thanks to support by IfU GmbH, the University of Cologne, the Met-Office, Laboratoire d'Aérologie, Meteoswiss, ONERA, and Radiometer Physics GmbH. Data are managed by the French national center for Atmospheric data and services AERIS. The Study of the water Vapour in the polar AtmosPhere (SVAAP) field campaign was supported by the Italian Antarctic research program (PNRA). The OCEANET-Atmosphere team of TROPOS around Ronny Engelmann and Dietrich Althausen are acknowledged for the acquisition and provision of HATPRO data aboard the Polarstern RV. EUREC4A is funded with support of the European Research Council (ERC), the Max Planck Society (MPG), the German Research Foundation (DFG), the German Meteorological Weather Service (DWD) and the German Aerospace Center (DLR). The work of Claudia Acquistapace was funded by the EXPATS research project (project number 4823IDEAP5) as part of the framework of the IDEA-S4S network in close collaboration with the Deutscher Wetterdienst (DWD), funded by the Federal Ministry for Digital and Transport (BMDV).

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
