# Peer review of "Atmospheric stability from numerical weather prediction"

_Atmospheric Measurement Techniques, 2024_

## Author Comment (AC1)

**RC1**: 'Comment on amt-2024-186', Anonymous Referee #1, 20 Dec 2024

A nice discussion / evaluation of wind park requirements, and an assessment of different microwave ground based observations, I have only some minor comments, shown below with Page Line:

We thank the reviewer for the positive feedback and useful suggestions. We received all the suggestions as detailed below (in red font).

- Figure 1: What is the difference between the top 2 plots (besides that they compare to different model data)? Are these different locations, thus the measurements are not the same? And might be good to include some general statistics in all the plots, as text (as done later in other plots, e.g. correlation).

The location is the same for the two plots: the HKZA floating lidar location. The ylabel for the plot on the right-hand side should read "number of half-hourly samples", as NEWA time series are provided stamped every 30-minutes. This explains the difference between the two "meas. via Rib" histograms. The ylabel and caption have been updated accordingly. The correlation coefficients are now shown within the scatter plot panels.

- Figure 2: the manuscript has top 1 plot, bottom 2, not left and right.

Thanks for spotting this typo. Left and right are now replaced with top and bottom, respectively.

- P9L216: The presented results show that global NWP / re-analysis models can be used, but that is unlikely to be valid globally. I assume ERA5 is particularly good in areas where a lot of measurements are available.

As the reviewer suggests, in some regions reanalysis dataset may fail to characterise key drivers such as the difference between air- and sea surface temperature. We added the following sentence towards the end of Section 2.1: "Conversely, care needs to be taken in regions where such main drivers (air- and sea surface temperature difference, for instance) may be incorrectly represented in NWP datasets, a validation of these quantities is always recommended (e.g., Section 4.2.2 of Borvarán et al, 2021)."

- Section 2.2: Is ECMWF actually assimilating these tower measurements? If yes, then they are not independent when doing validations.

We confirm that tower measurements are not assimilated into NWP models and thus they are independent from each other. This is now clarified in the revised manuscript (Section 2.2).

- P9L226: Are these 2 models providing data exactly on the vertical levels of the tower? Or better/worse? Any averaging required to align the resolutions?

NEWA air temperature data are provided at 2, 50, 75 and 100m. DOWA data are provided at 10, 20, 40, 60, 80, 100m. Model data have been interpolated at the measurement elevations. This information is now provided in Section 22. No averaging is applied.

- Figure 5: Might be more instructive to use a range that is within the atmospheric conditions (I assume, -100K change over 1km is not really a valid range), and print the lengthy legend on the side of the plot?
  Also, are you doing some outlier detection and removal? Sometimes the model seems to be "stuck" at about -20K/km, while the tower detects values up to 150K/km.

Agreed. Figure 5 has been modified to reduce the legend and the axis range to [-50 150] K/km. We confirm we have not applied data screening such as outlier detection and removal. However, the models (NEWA or DOWA) do seem unable to capture temperature differences larger than approximately -0.5°C between the model output at 100 and 50 m levels (corresponding to -20K/km), e.g., the minimum value is -0.6°C for the entire NEWA time series. This may be due to the treatment of surface stability in these models, but we have not been able to identify the root cause for these. Such instances occur for very unstable conditions with very small wind speeds (less than 4 m/s), i.e. not significant for the most engineering analyses discussed in the paper (wind turbines have cut-in wind speeds around 4 m/s). We added this consideration to Section 2.2.

- Figure 6: (a) I assume the figures should somewhere have the a, b, c, d label?; (b) the map plot should have lat/lon labels; (c) the lower left plots have no y label units.

Agreed. Missing panel and axis labels have been added. Thanks!

- Figure 7: Again, the a, b, c, d labels are missing, making it difficult to understand this already complex plot compilation.

Agreed. Labels have been added to panels. Thanks!

- P22L516: Just interpolation to the radiosonde data? The resolution of MW is much lower, thus you could also average the sonde data (or fold it with an averaging kernel of the MW).

We agree that the resolution of MWR is much lower than that of radiosonde data, and that sometimes the MWR retrieved profiles are validated against radiosonde data smoothed according to the MWR averaging kernels (e.g., Löhnert and Maier, 2012). This step brings the radiosonde profile onto the vertical resolution of the MWR retrieval, and it generally improves the statistical agreement of the two profile sources. However, this analysis aims to quantify the performances of MWR retrievals to catch the vertical gradient between two fixed heights, such as measured by radiosonde or tower sensors, for the interest of wind energy applications, with no special processing with respect to what is available from the MWR manufacturer. We added the above considerations in Section 3.3 (near former line 516).

- Figure 9: Is there a possibility to split this into stable/unstable atmospheric conditions?

Following Eq.(1) and the related discussion in Section 1 (Introduction), we identify stable/unstable conditions with the sign of the potential temperature gradient. Thus, stable/unstable conditions are separated by the horizontal line at 0 K/km. We modified Figure 9 caption to make this clear.

- P23L540: Resolve a.g.l. somewhere.

Thanks! We changed to asl to be consistent throughout the manuscript.

- P24L552: Sorry, I might have missed it above, but how are MW data sets averaged in time to the sonde times? And they provide different scanning modes, as indicated above. Does that influence the averaging? And are you considering the ascent times, or just an average time of the sonde?

Temporal colocation between MWR measurements and radiosonde data is achieved averaging the MWR measurements within 30 minutes after the radiosonde launch (as stated at lines 514-515 of the original manuscript). We only consider radiosonde data during the ascent flights. The scanning mode is different from one dataset to the other, but it does not influence the averaging since zenith-only and scanning mode retrievals (where both available) are treated separately. This is now clearly stated in Section 3.3. For example, Figure 13 compares results for zenith-only and scanning mode retrievals for the two datasets where these are both available (POL, MET). The scanning mode used for the retrievals is now indicated in Table 2 and Section 3.2.

- Figure 11: Is there a correlation difference between stable and unstable conditions?

As recalled above, we identify stable/unstable conditions with the sign of the potential temperature gradient. Thus, stable/unstable conditions are separated by the horizontal/vertical line at 0 K/km. We modified Figure 11 caption to make this clear. For this ENA dataset, correlation is higher in unstable (UNS) than in stable (STB) conditions (0.6 vs. 0.45, see right panel below). However, it is the opposite for the MOL dataset (0.39 vs. 0.9, left panel below), likely because of the different spanned range. Therefore, we prefer to avoid conclusive statements.

[Figure]

- P25L588: Any comment regarding the different seasons being used? And with 4 sondes, you'll also capture different diurnal variations, compared to the 2 sondes at GRA.

Agreed. We added the following comment to Section 4 (Validation): Note that the range of potential temperature gradients is larger in MOL than ENA datasets, due to the combination

of larger sample, different environment and season, and finally better coverage of diurnal cycle (4 vs. 2 daily radiosondes).

- Figure 12: Maybe be consistent in using titles of the plots. Regarding panel a, the agreement is remarkable between 300m to 700m. Is there any use of sonde data in the MW retrieval?

Agreed. Titles have been removed from all panels. Labels have been added to indicate dataset and instrument. Also, the same x- and y-axis ranges are now used for all panels. We agree that the bias is remarkably small between 300-700 m. We confirm that the radiosonde data considered for the comparison are not used in the MW retrieval, which is trained with historical radiosonde data.

- Figure 13: Maybe also include titles? No, after further reading, please use titles, so that the campaign/site is easy identifiable. And regarding presentation, maybe make a blank plot c, write no data into it?

Agreed. Labels have been added to indicate dataset and instrument. Also, the same x- and y-axis range is now used for all panels to facilitate the comparison. Data in the former panel (b) are now divided into panels (b) and (c), so as to have panel-by-panel correspondence with Figure 12. Thus, panel (c) is no longer blank.

- P29L622: While reading this: are these MW retrievals all consistent? Or each uses its own retrieval setup? If the setups are different, what is the impact on the assessed performance and uncertainty?

Retrieval techniques used to produce different datasets are different. Information on the retrieval method used within each dataset has been added to the revised manuscript (e.g., Table 2 and Section 3.2). The retrieval technique, as well as instrument calibration, do impact the assessed performance. This analysis relies on the retrieval technique and calibration procedure applied by the data provider, and thus it provides an indication of the typical performances to be expected by commercial MWR units, without any special post-processing. This information has been added at the end of Section 3.2.

- P30L647: I think this is the first time, a quality control is mentioned. What does it consist of, and is it used for other data sets too (see also comment above)?

This quality control is unique to this dataset. It was deemed necessary to purge unrealistic retrievals found in the first 2-week period of the 2007 cruise (20 April to 3 May). These temperature retrievals were characterised by a suspicious large nose at 1 km height (~15 K contrast), the cause of which was not found. This information is now given in Section 4.

- P30L663: How robust is this improve correlation? There is one data point at about 30K/km for scanning. Do you still get good correlation if that data point is removed?

As stated in the original manuscript (at line 665), we agree that the improved correlation is mostly driven by only one point (at 27 K/km). If that point is removed, then the correlation would not be improved substantially. However, we decided to leave it in, as that point corresponds to the only remaining case in very stable conditions where both zenith and

scanning retrievals are available, supporting theoretical expectations that scanning shall improve the capability to catch vertical gradients.

- P30L668: Okay, here is finally some remark on the different retrieval methods. As these seem to vary quite a bit, I think they should be shortly mentioned, maybe in the section where the instruments are introduced.

Agreed. Information on the retrieval method and the a-priori used within each dataset have been added to the revised manuscript (e.g., Table 2 and Section 3.2).

- P31L693: Not for this work, but it would be a very good follow on work, to assess these different instruments / locations / seasons with the same retrieval algorithm. -> saw you identified the need in the conclusion. Good.

Agreed.

- Table 3: Here AGL is used. Is that introduced somewhere?

Thanks! We changed to ASL to be consistent throughout the manuscript.

- P32L703: While reading these different gradients - is there actually a limit what gradient can be detected by which instrument, as they do differ in observation capability, e.g. such as the number of channels, polarisation, etc?

The vertical resolution of MWR temperature profile retrievals (and thus the ability to accurately quantify potential temperature gradients) does depend on a number of factors, including the number of channels, the channel frequencies, the channel sensitivity and beamwidth, and the scanning procedure. It also depends on data processing, i.e., retrieval method and a priori information. Since these factors change from one dataset to another, this study cannot conclude about limitations of one instrument or the other. Here we can only conclude that the considered datasets indicate that MWR in general can detect potential temperature gradients at least from -5 to +50 K/km. This consideration is now stated in Section 5.

---

## Author Comment (AC2)

**RC2**: 'Comment on amt-2024-186', Anonymous Referee #2, 24 Jan 2025

**Review for AMT-2024-186**

Title: "Atmospheric stability from microwave radiometer observations for on/offshore wind energy applications"

Authors: Domenico Cimini, Rémi Gandoin, Stephanie Fiedler, Claudia Acquistapace, Andrea Balotti, Sabrina Gentile, Edoardo Geraldi, Christine Knist, Pauline Martinet, Saverio T. Nilo, Giandomenico Pace, Bernhard Pospichal, Filomena Romano

**General comments:**

The study presented in this manuscript first reviews stability metrics derived by NWP models and reanalysis, useful for wind energy. Second, it quantifies the performances of different microwave radiometers at estimating stability metrics on land and offshore. The manuscript is relevant, well written and easy to read. I think it fits well in the scope of AMT. I suggest only some minor modifications and to include some additional information. For instance, some more info on retrievals' and calibration's procedures, a-priori or radiosonde dataset used in the retrievals, radiosonde type, should be provided.

We thank the reviewer for the positive feedback and useful suggestions. We received all the suggestions as detailed below (in red font).

The title of the manuscript does not mention the first part of the study. The authors might want to consider if to mention this as well.

Agreed, thanks for the suggestion. The title now reads: "Atmospheric stability from numerical weather prediction models and microwave radiometer observations for on/offshore wind energy applications"

**Specific comments:**

Page 1, Abstract, line 32: Include units for RMS.

Agreed. Thanks!

Page 5, line 138: "Measurement data came from the FINO1, FINO2 and FINO3 met masts". How are these data used in the before mentioned NWP models? Are they assimilated?

We confirm that tower measurements are not assimilated into NWP models and thus they are independent from each other. This is now clarified in the revised manuscript (Section 2.2).

Page 6, Figure 1 caption: Describe the pink line in the caption of the figure. Also, in figures with multiple panels, a), b) c)… would be useful.

Agreed. Thanks!

Page 7, line 194: How id the boundary-layer height determined during stable vs unstable conditions?

The boundary layer height is parameterized using friction velocity and latitude, as proposed by Gryning et al. (2007), independently of atmospheric stability. This is now explicitly stated in Section 2.1.

Page 8, Figure 2 caption: Panels are 'top' and 'bottom', not 'left' and 'right'. Also, in the caption you say '(8 and 12 m/s, 207 respectively)', but in the title of the panels you say '4 +/- 0.5' and 8 +/- 0.5'.

Agreed. Figure caption has been revised accordingly. Thanks!

Page 11, Figure 4 caption: This is another figure with multiple panels, where a), b) c)… would be useful.

Agreed. Figure 4 panels have been labelled.

Page 11, Figure 5: Do the models have hard limits on e lower values of dT/dz?

We confirm we have not applied hard limits to lower dT/dz values. However, the models (NEWA or DOWA) do seem unable to capture temperature differences larger than approximately -0.5°C between the model output at 100 and 50 m levels (corresponding to -20K/km), e.g., the minimum value is -0.6°C for the entire NEWA time series. This may be due to the treatment of surface stability in these models, but we have not been able to identify the root cause for these. Such instances occur for very unstable conditions with very small wind speeds (less than 4 m/s), i.e. not significant for the most engineering analyses discussed in the paper (wind turbines have cut-in wind speeds around 4 m/s). We added this consideration to Section 2.2.

Page 13, Figure 6: Please include a), b) c)… to the panels. Also, include x- and y-labels to the upper left panel, and y-labels to the lower two left panels.

Agreed. Missing panel and axis labels have been added. Thanks!

Page 14, Figure 7: Please include a), b) c)… to the panels. This figure really needs them (same for all the figures in the Supplemental material). For the figures in the supplemental material, it is interesting to see the jumpy behavior of the zi time series, particularly for the FINO1 site. This is why it might be interesting to know how are these zi values obtained.

Agreed. Panels have been labelled. Here, the ABL height comes from model (NEWA). The jumpy behavior in the time series may be due to the way the underlying NWP model (WRF MYNN) computes the ABL height, i.e by selecting a discrete model level (as seen in the subroutine GETPBLH within the following fortran module: https://github.com/wrf-model/WRF/blob/master/phys/module_bl_mynn.F).

Page 16, Table 1: I think it'd be useful to list somewhere the retrieval techniques utilized for each of the MWRs? Also, what a-priori were used?

Agreed. Information on the retrieval method and the a-priori used within each dataset have been added to the revised manuscript (e.g., Table 2 and Section 3.2).

Page 17, after line 368: Could you describe what type/schedule of maintenance, calibration were performed on these MWRs?

Agreed. Information on MWR calibration/maintenance have been added to the revised manuscript (Section 3.2).

Page 19, line 435: How many daily radiosondes per day and at what time?

ARM launches two daily radiosondes from ENA at 11:30 and 23:30 UTC. This is now stated in Section 3.2.

Page 20, line 473: What type of MWRs?

The MWR is a 14-channel RPG HATPRO. This is now stated while introducing THAAO. Thanks for noticing we missed that!

Page 23, Section 4: Would it make more sense to add the info on Radiosonde/MWR matchups, radiosonde launch times, radiosonde type, and so on in a table, rater than in the text for each dataset? In this way, you could also remove some of these details from the individual dataset descriptions in Section 3.2.

Agreed. Information on radiosonde/MWR matchups and radiosonde launch times has been added to Table 2.

Page 30, line 655: 'mostly driven by only one point'. I agree.

As noted by the reviewer, we admit that the improved correlation is mostly driven by only one point (at 27 K/km). If that point is removed, then the correlation would not be improved substantially. However, we decided to leave it in, as it is the only remaining case in very stable conditions where both zenith and scanning retrievals are available, supporting theoretical expectations that scanning shall provide more accurate vertical gradients.

Page 32, lines 710-711: No mention to the different method used for the retrievals (and the a-priori used for these) is given up to this point, nor about calibration performed, but I think it would be an useful information to add to the Section with the different dataset descriptions. Good calibrations and adequate a-priori datasets are crucial for MWRs.

We concur that adequate a-priori information and calibration procedures are crucial for accurate MWR retrievals. In this analysis we rely on the calibration procedures and quality control applied by the data provider. These are trustable national weather services (e.g., DWD, MeteoFrance), scientific programmes (ARM), or research campaigns (SOFOG3D, SVAAP, EUREC$^4$A). This information has been added at the end of Section 3.2. In addition, information on retrieval methods, a-priori, and calibration/maintenance have been added to the revised manuscript (e.g., Table 2 and Section 3.2).